# Trickle infection and immunity to *Trichuris muris*

**Maya Glover** [1,2‡], **Stefano A. P. Colombo** [1,2‡], **David J. Thornton** [1,2], **Richard K. Grencis** [1,2] *

**1** Lydia Becker Institute of Immunology and Inflammation, University of Manchester, Manchester, United Kingdom, **2** Wellcome Centre for Cell Matrix Research, University of Manchester, Manchester, United Kingdom

‡ These authors share first authorship on this work.
* richard.grencis@manchester.ac.uk

**Data Availability Statement:** All relevant data are within the manuscript and its Supporting Information files.

**Funding:** This work was funded by Wellcome Trust Studentships (Grant number 103132/Z/13/Z [MG

## Abstract

The majority of experiments investigating the immune response to gastrointestinal helminth infection use a single bolus infection. However, *in situ* individuals are repeatedly infected with low doses. Therefore, to model natural infection, mice were repeatedly infected (trickle infection) with low doses of *Trichuris muris*. Trickle infection resulted in the slow acquisition of immunity reflected by a gradual increase in worm burden followed by partial expulsion. Flow cytometry revealed that the CD4+ T cell response shifted from Th1 dominated to Th2 dominated, which coincided with an increase in Type 2 cytokines. The development of resistance following trickle infection was associated with increased worm expulsion effector mechanisms including goblet cell hyperplasia, Muc5ac production and increased epithelial cell turn over. Depletion of CD4+ T cells reversed resistance confirming their importance in protective immunity following trickle infection. In contrast, depletion of group 2 innate lymphoid cells did not alter protective immunity. *T. muris* trickle infection resulted in a dysbiotic mircrobiota which began to recover alpha diversity following the development of resistance. These data establish trickle infection as a robust and informative model for analysis of immunity to chronic intestinal helminth infection more akin to that observed under natural infection conditions and confirms the importance of CD4+ T cell adaptive immunity in host protection.

## Author summary

Infection with parasitic worms (helminths) is a considerable cause of morbidity in humans. Understanding how we respond to infection is crucial to developing novel therapies. Laboratory models of helminth infection have been a valuable tool in understanding fundamental immune responses to infection. However, typically an individual mouse will be infected with a large, single-dose of the parasite. This is in contrast to the natural scenario in which individuals will receive frequent low level exposures. However, it is unknown how repeated infection alters the development of immunity to infection. We have developed a laboratory model to tackle this question. We infected mice with the

& SAPC]; https://wellcome.ac.uk/funding/schemes/
doctoral-studentships) and Wellcome Trust
investigator award (Z10661/Z/18/Z [RKG]; https://
wellcome.ac.uk/funding/schemes/investigator-
awards-science). The Wellcome Trust Centre for
Cell Matrix Research, University of Manchester is
supported by centre funding from the Wellcome
Trust (203128/Z/16/Z [DJT]; https://wellcome.ac.
uk/news/wellcome-centre-awards). The funders
had no role in study design, data collection and
analysis, decision to publish, or preparation of the
manuscript.

**Competing interests:** The authors have declared
that no competing interests exist.

model helminth *Trichuris muris* on a weekly basis and assessed a range of responses in
comparison with a more traditional infection regime. We found striking differences in the
dynamics of the infection, the host immune response, and in changes to host gut micro-
bial populations. Our study shows how resistance to helminth infection can develop over
time in response to repeat infection, and provides a model system that better reflects
human immunity to this parasite.

## Introduction

Gastrointestinal (GI) dwelling nematodes infect approximately 1 billion people worldwide
causing significant ill health[1]. Prevalence is high in endemic areas although intensity of
infection varies with age, suggesting acquired immunity develops, although sterile immunity is
rare and individuals are repeatedly challenged with low numbers of infectious stages through-
out their lives [2]. Thus, identifying mechanisms of acquired immunity in human populations
is extremely challenging, however, data is supportive of acquired immunity driven by Type 2
immunity for at least some if not all the major soil transmitted helminths (STH) *Ascaris lum-
bricoides, Necator americanus, Ancylostoma duodenale* and *Trichuris trichiura* [3]. Despite this,
immunity appears to be only partial at best and takes considerable time to develop [2,4].

Animal models of intestinal nematode infection have been widely used to help define mech-
anisms of immunity to these types of pathogen [5]. Typically, rodents are given a single, and
sometimes a second, infection composed of an unnaturally large dose of infectious stages. This
approach has been extremely informative and established that such a robust parasite challenge
stimulates the host to generate host protective responses, dominated by Type 2 immunity. A
clear role for CD4+ Th2 cells is well documented but in addition, recent data indicates that
innate immunity plays a major role in host protection through tuft cell induced Type 2 innate
lymphoid cell (ILC2) production of IL-13 [6–8]. Whilst single bolus infections in the labora-
tory have been central to defining paradigms of resistance and susceptibility, one major dis-
crepancy between these models and natural infection in man, and indeed rodents, is that
individuals are infected repeatedly with low dose infections throughout their lifetime and STH
are chronic, non-resolving infections. Studies of repeated low dose nematode infections of
rodents are indeed rare [9,10].

The mouse whipworm, *Trichuris muris*, is uniquely placed as a model of the human STH
*Trichuris trichiura*, in that it exists as a natural chronic infection of wild rodents. The antigenic
cross-reactivity of the two species, similar morphology and same niche of intestinal infection
establishes *T. muris* as a suitable parasite to model human infection and has been consolidated
by the recent description of the *T. trichiura* and *T. muris* genomes [11,12]. Moreover, the
immune responses required for resistance and susceptibility to *T. muris* are well established
and influenced by both infective dose and strain of laboratory mouse. C57BL/6 mice that
receive a single high dose infection develop resistance, dependent on CD4+ Th2 cells [13] and
IL-13 production [14]. IL-13 induces a number of effector mechanisms that mediate worm
expulsion including accelerated epithelial cell turnover [14–16]. *Trichuris* embed in the epithe-
lium of the caecum and with increased epithelial cell turnover the worm is physically carried
out of the epithelium into the lumen and is expelled. In resistant mice, IL-13 also induces gob-
let cell hyperplasia and elevated Muc2 and Muc5ac mucins, with deficiency of these mucins
causing susceptibility to infection [17].

C57BL/6 mice that receive a single low dose infection of *T. muris* are, in contrast, suscepti-
ble and develop a chronic infection associated with CD4+ Th1 cells, IFN-γ production [18]

and subsequent regulation via IL-10 [19,20]. Chronic low level *Trichuris* infection is associated with a dysbiosis of the intestinal microbiota with a reduction in microbial diversity [21,22] that has a survival benefit for the parasite [23].

Here, we have established a natural trickle infection with *T. muris* in the mouse using repeated low doses of eggs to mimic more closely exposure in the field. Following comprehensive immune phenotyping, we now show that repeated low doses of *T. muris* infection results in the slow development of partial resistance with the modification of a Type 1 dominated response to a functionally protective Type 2 immune response. Resistance after this infection regime is dependent on CD4+ T cells and associated with Muc5ac production and an increase in intestinal epithelial cell turnover, confirming their importance even after this multi-infection regime. ILC2s, however, did not appear to play an important role in the development of resistance following *T. muris* trickle infection or indeed after a single high dose infection, which induces complete Type 2 cytokine mediated parasite clearance.

We propose that this infection regime presents a powerful approach to further dissect the host/parasite relationship of a major neglected tropic disease, trichuriasis, using a highly relevant mouse model under conditions where the host is challenged by parasite infection in a manner more akin to that seen under natural infecting conditions. Going forward, this provides a more representative platform for analysis of potential vaccine candidates and novel anti-helminthics.

## Results

### Development of resistance and Type 2 immune responses following *T. muris* trickle infection

To replicate a more natural infection regime of *T. muris*, mice were infected weekly with low doses of eggs (20 eggs) for 3, 5, 7 or 9 weeks (S1 Fig). Two weeks after the final infection worm burden and immune responses were analyzed. Total worm burden revealed *Trichuris* worm numbers increased through weeks 5 to 9 post infection (p.i.) followed by a significant decrease by week 11 (Fig 1A). Faecal egg counts at week 11 followed a similar pattern and suggested a significant reduction in worm burden occurs after week 9 (Fig 1B). When *Trichuris* eggs hatch in the caecum, L1 larvae are released and undergo a series of moults, L2-L4, before developing into adult worms. Larval stages were counted and revealed an absence of early larval stages at week 11 (Fig 1C). Analysis of CD4+ T helper cell subsets from large intestinal lamina propria revealed a shift from a Th1 dominated response during susceptibility at week 9, to a Th2 dominated immune response, associated with resistance at week 11 (Fig 1D). The peak in Th2 cells coincided with an increased capacity to produce IL-13 and no significant change in IFN-γ production (Fig 1E). Analysis of CD4+ T cell subpopulations from MLNs did not show any significant changes following *T. muris* trickle infection during the period of resistance (S2 Fig).

Previous work has demonstrated that antibody responses are not required for resistance to single *T. muris* infections [18,19]. Antibody responses specific for *T. muris* adult and larval antigens were analysed following trickle infection. Type 1 IgG2a/c antibody and Type 2 IgG1 antibody steadily increased throughout the complete infection time course (S3 Fig). No distinct difference in antibody responses against each stage was apparent (excluding responses against L1 antigens which remained low throughout trickle infection). Additionally, total IgE levels did not significantly increase throughout trickle infection (S3 Fig).

Goblet cell number and goblet cell size increased consistently throughout the course of the trickle regime peaking at week 11 when resistance had been established (Fig 2A–2C). Crypt length, an indicator of gut inflammation, increased during trickle infection (Fig 2D). Sections were stained with HID-AB to visualize changes in mucin glycosylation in the caecum (Fig 2E).

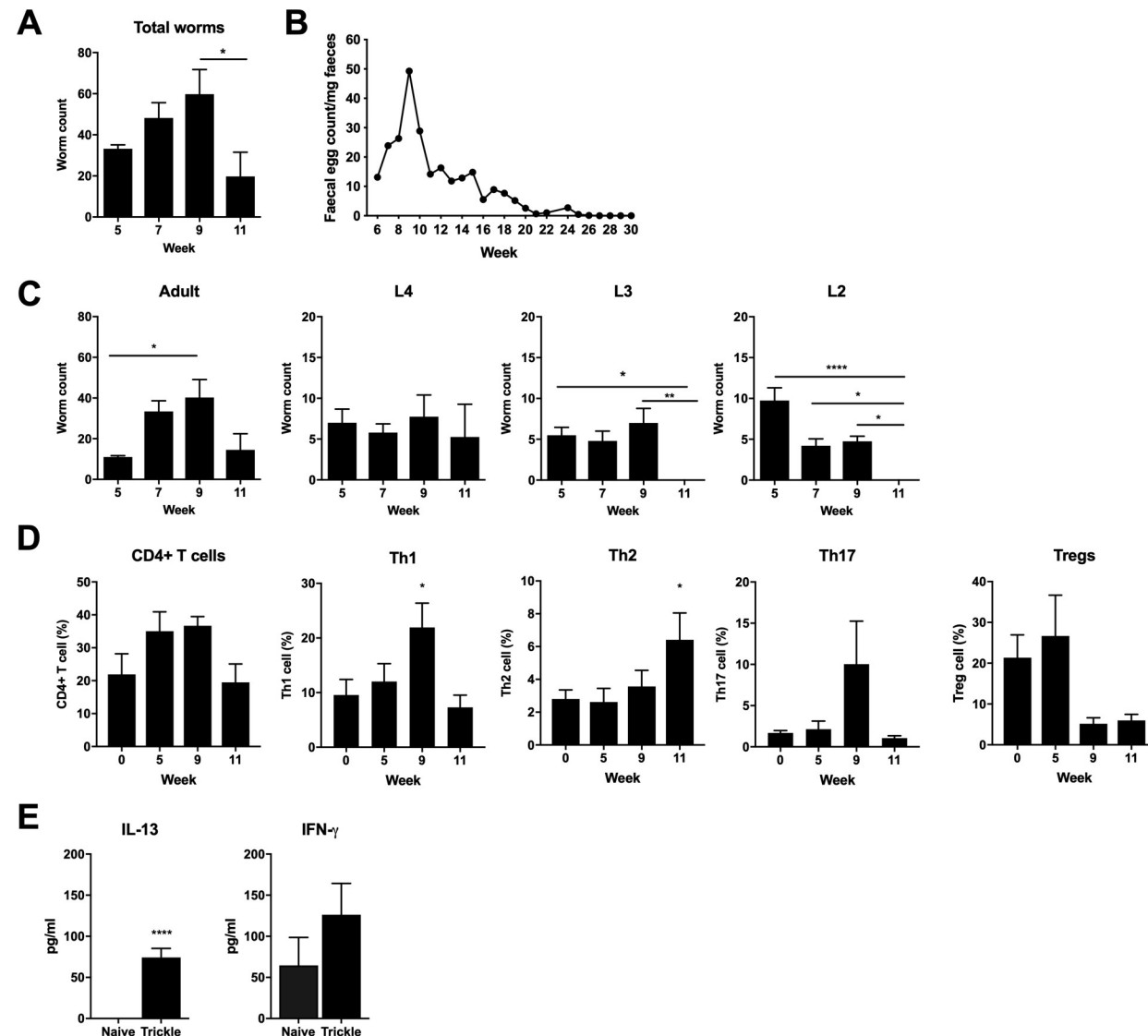

**Fig 1. Development of resistance following *T. muris* trickle infection.** C57BL/6 mice were infected repeatedly with low doses of *T. muris*. A) Total worm burdens measured by eye under dissecting microscope. B) Faecal egg counts of mice trickled for 9 weeks were tracked from 6 weeks post the primary infection until eggs could no longer be detected in the faeces. Egg burden was quantified using a McMaster count chamber. Points represent the mean of 5 individual mice. C) Adult and larval *Trichuris* worm burdens, n = 5. D) CD4+ T cell percentage from large intestinal lamina propria, Th1 (Tbet+), Th2 (GATA3+), Th17 (RORγt+), Tregs (FOXP3+), n = 10, from two independent experiments. CD4+ T cell percentage calculated as percentage of all live cells. T cell subset percentage calculated as percentage of all CD4+ cells. E) Cytokine production from MLN cells collected at 11 weeks p.i. and re-stimulated *ex vivo*. Levels of secreted cytokine in the supernatant were quantified by cytometric bead array, n = 5. Statistical analysis completed by one way- ANOVA or unpaired t test. Data presented as mean +/- SEM. * = p<0.05, ** = p<0.01, **** = p< 0.0001. Representative data from 2 independent experiments.

Following low dose *T. muris* infection mucins become sialylated and are more vulnerable to degradation by *T. muris* products [17,24]. However, following high dose *T. muris* infection mucins stay sulphated [20]. During *T. muris* trickle infection mucins appeared sulphated throughout.

Goblet cells produce a number of protective mucins and secreted proteins during resistance to *T. muris* [17]. To determine which goblet cell proteins were correlated with resistance during trickle infection, the expression of a number of goblet cell associated genes was assessed.

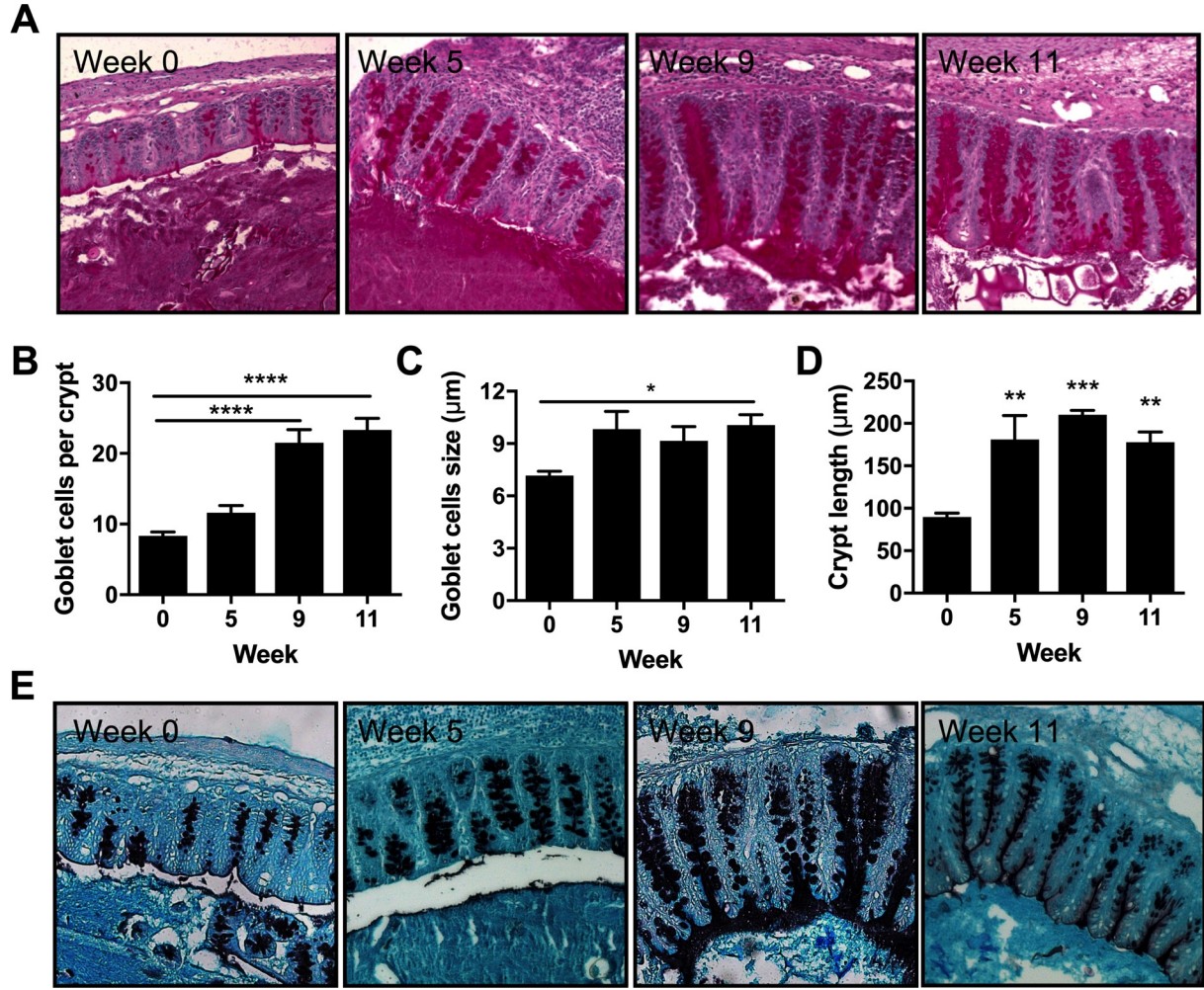

**Fig 2. Goblet cell responses in *T. muris* trickle infected mice.** Paraffin embedded caecal sections from trickle infected mice were PAS stained and goblet cells and crypt length were analyzed using ImageJ. (A) Representative imagines of PAS stained caecal sections. (B-D) Individual quantification for each mouse was achieved by measuring at least 10 individual crypts and determining the mean number of (B) goblet cells per crypt, (C) goblet cell diameter, and (D) Crypt length. Bars represent the mean across 5 mice. (E) Representative imagines of HID-AB caecal sections to visualize sulphated (black) and sialylated (blue) mucins. Statistical analysis completed by a one way-ANOVA. Data presented as mean +/- SEM. * = p<0.05, ** = p<0.01, *** = p<0.001, **** = p< 0.0001.

There was a specific increase in Muc5ac expression at week 11 after trickle infection when resistance developed (Fig 3A). The increase in Muc5ac was confirmed by western blot on secreted mucus. The majority of trickle infected mice showed an increase in secreted Muc5ac (Fig 3B). Muc2 is also associated with resistance in *T. muris* models of infection [25]. Despite no significant increase in gene expression, staining of caecal sections revealed an increase in Muc2 positive cells during the development of resistance in trickle infection (Fig 3C). Relm-β (and to a lesser extent TFF3) have also been associated with resistance in *T. muris* infection [26]; despite no increase in TFF3, Relm-β expression significantly increased during the generation of resistance in trickle infection (Fig 3A).

An increase in epithelial cell turnover occurs in resistance of *T. muris* following a high dose infection and is hypothesized to physically move the parasite out of its optimal intestinal niche resulting in expulsion [16]. To determine whether increased turnover was induced during resistance following trickle infection, mice were injected with BrdU to assess cell turnover

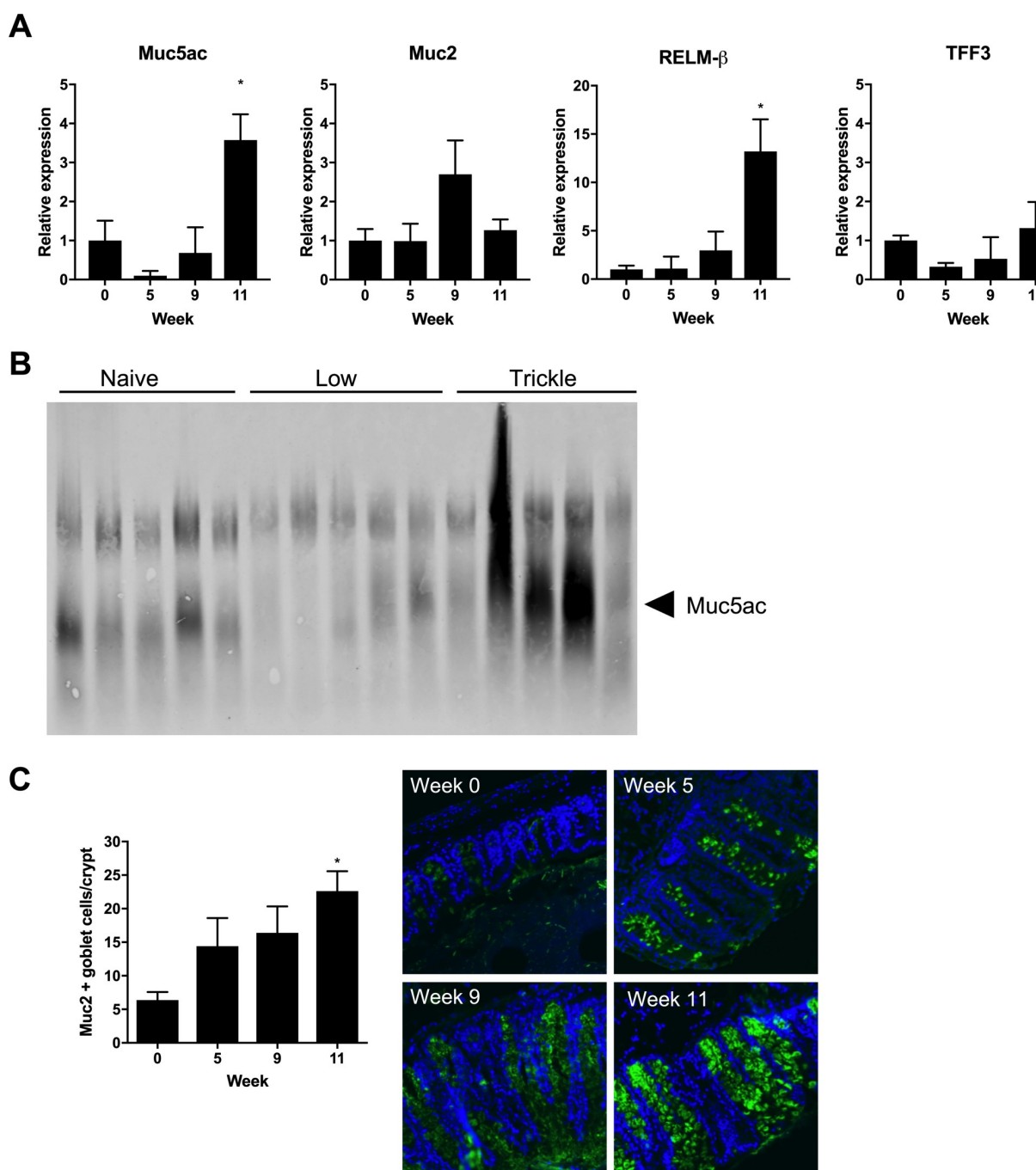

**Fig 3. Mucin production in *T. muris* trickle infection.** (A) Relative expression of goblet cell-associated genes measured by qPCR. Relative quantification was achieved using the $2^{-\Delta\Delta CT}$ method with β-actin used as the housekeeping reference gene. (B) Western blot to visualize Muc5ac protein from extracted mucus from naïve or *T. muris* infected C57BL/6 mice at 11 weeks p.i.. (C) Quantification of the mean number of Muc2+ goblet cells per crypt, given as the mean number of Muc2+ cells from a minimum of 10 crypts per mouse. Representative immunostained sections during trickle infection. Muc2 in green. DAPI stain in blue. n = 5, statistical analysis completed by a one way-ANOVA comparing each time point to week 0. Data presented as mean +/- SEM, * = p<0.05.

[27]. A significant increase in epithelial cell turnover was observed during the generation of resistance at week 11 of trickle infection (Fig 4A and 4B) although, no increase in amphiregulin gene expression was observed (Fig 4C) [27].

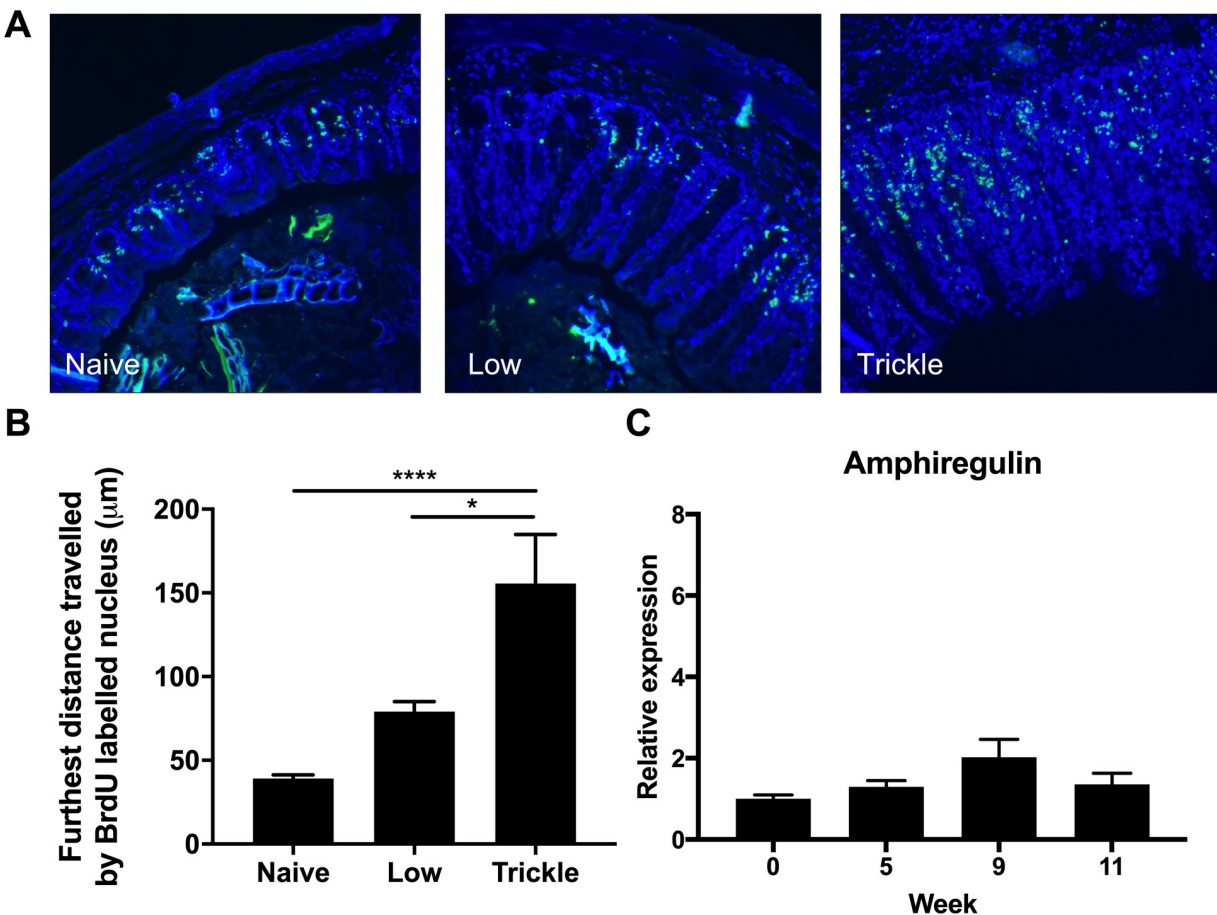

**Fig 4. Epithelial cell turnover following *T. muris* trickle infection.** At week 11 following either low dose infection (20 eggs) or a 9 week trickle infection (20 eggs/week), 9 hours prior to sacrifice, mice were I.P. injected with BrdU to identify proliferating cells. (A) Caecal sections were stained for anti-BrdU (green) and DAPI (blue) and the distance the furthest BrdU stained cell was measured. (B) Quantification of epithelial turnover was determined by measuring the distance between the base of each crypt and the furthest BrdU+ve cell from that point. A mean value was generated for each mouse from at least 10 separately analyzed crypts. (C) Relative expression of amphiregulin in the caecum measured by qPCR. Relative expression was calculated using the 2^-ΔΔCT method with β-actin used as a housekeeping reference gene. n = 5, statistical analysis completed by a one way-ANOVA or unpaired t test. Data presented as mean +/- SEM, ** = p<0.01.

### Resistance developed following *T. muris* trickle infection is long lasting and protects against subsequent infection

To determine whether the resistance that developed following *T. muris* trickle infection was maintained even in the absence of active infection, trickle infected mice were challenged following either the natural eventual loss of worms or following anti-helminthic treatment. Faecal egg counts were assessed to determine when all worms had been expelled and the animals were then challenged at week 30. After a single low dose infection, mice were still susceptible to a challenge low dose infection whereas mice infected with single high dose were able to expel a challenge low dose infection. Trickle infected mice were protected against subsequent infection, similar to high dose infected mice (S4A Fig). Moreover, mice that had already received a short trickle infection (3 low dose infections) were not resistant to a low dose challenge given 10 weeks later, suggesting that the number of low dose infection events is key to generating resistance and not simply the slow development of protective immunity (S4B Fig).

## CD4+ T cells are essential for the development of resistance following trickle infection

To determine whether innate or adaptive cells were responsible for the development of resistance following trickle infection, RAG-/- mice were infected with repeated low doses following the week 11 infection regime. Although C57BL/6 mice developed resistance at this time point, RAG-/- mice did not expel worms, steadily building up worm numbers, confirming adaptive immune responses were essential for resistance (Fig 5A). To define a role for CD4+ T cells in the development of trickle infection-induced resistance, C57BL/6 mice were treated with anti-CD4 antibody between week 8 and 11, when we observe resistance developing. Depletion of CD4+ T cells was confirmed by flow cytometry (Fig 5B). This reduction resulted in a significant increase in total worm burden at week 11 which included adult worms and all larval stages (Fig 5C and 5D). Furthermore, depletion of CD4+ T cells resulted in a significant reduction in associated effector mechanisms including goblet cell hyperplasia, Muc5ac and Relm-β expression and epithelial cell turnover (Fig 6). There were small reductions in the *T. muris* specific antibody response following CD4+ T cell depletion (S5A and S5B Fig).

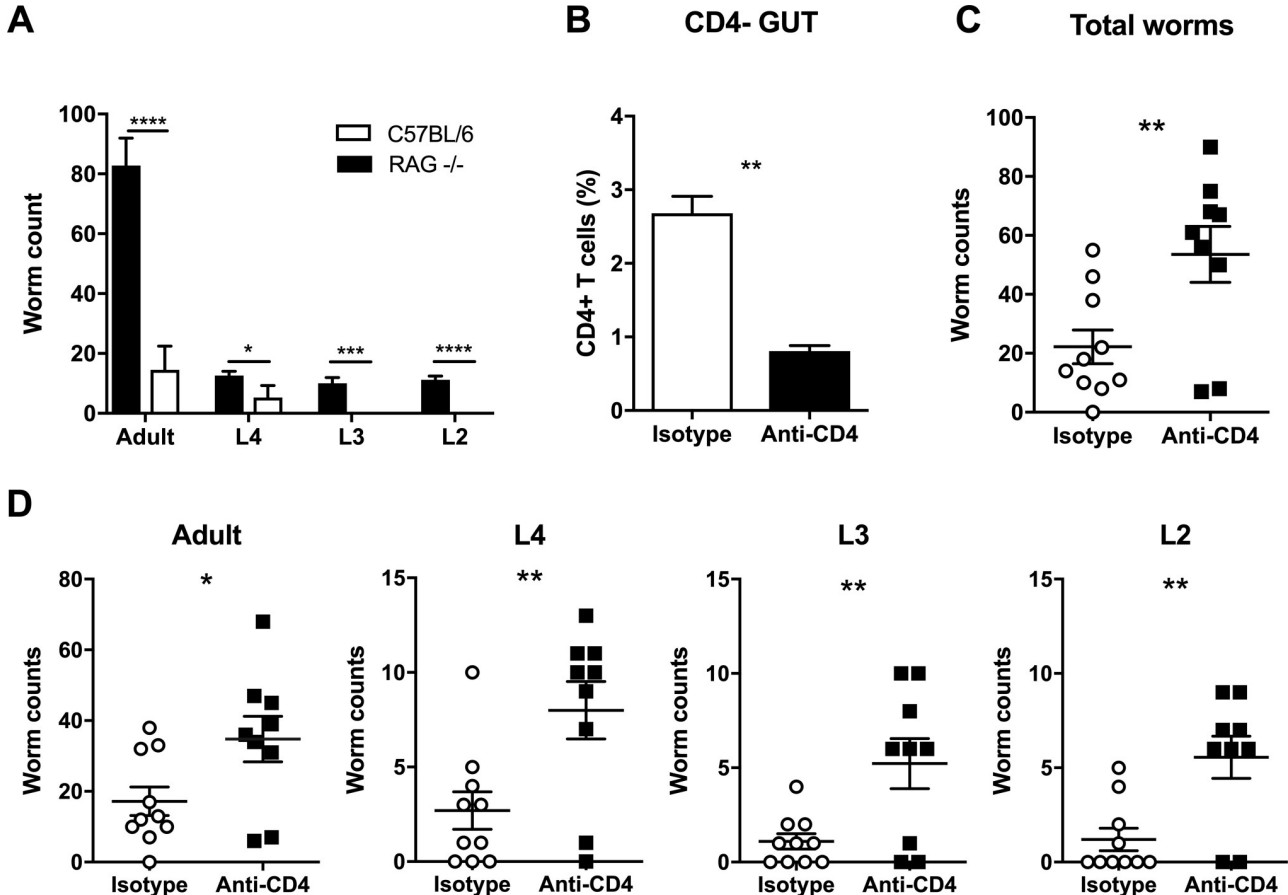

**Fig 5. CD4+ depletion in *T. muris* trickle infection.** (A) Worm burdens of *T. muris* trickle infected RAG-/- (black) and C57BL/6 (grey) mice following a 9 week trickle infection and sacrificed at week 11 post infection, n = 5. Adult worms and larval stages 4–2 were counted. (B-D) To determine the importance of CD4+ cells in generating immunity during trickle infection mice were treated with 200μg anti-CD4 antibody or isotype control antibody, 3 times a week for 3 weeks between weeks 8–10 of trickle infection. (B) FACS quantification of CD4+ T cells in the large intestine to confirm depletion, values are given as a percentage of live cells. (C) Total worm burdens of *T. muris* infected mice determined by dissecting microscopy. (D) Adult worms and larval stage counts following CD4+ depletion, n = 9–10, based on two experiments. Statistical analysis completed using an unpaired t test. Data presented as mean +/- SEM, * = p<0.05, ** = p<0.01, *** = p<0.001, **** = p< 0.0001.

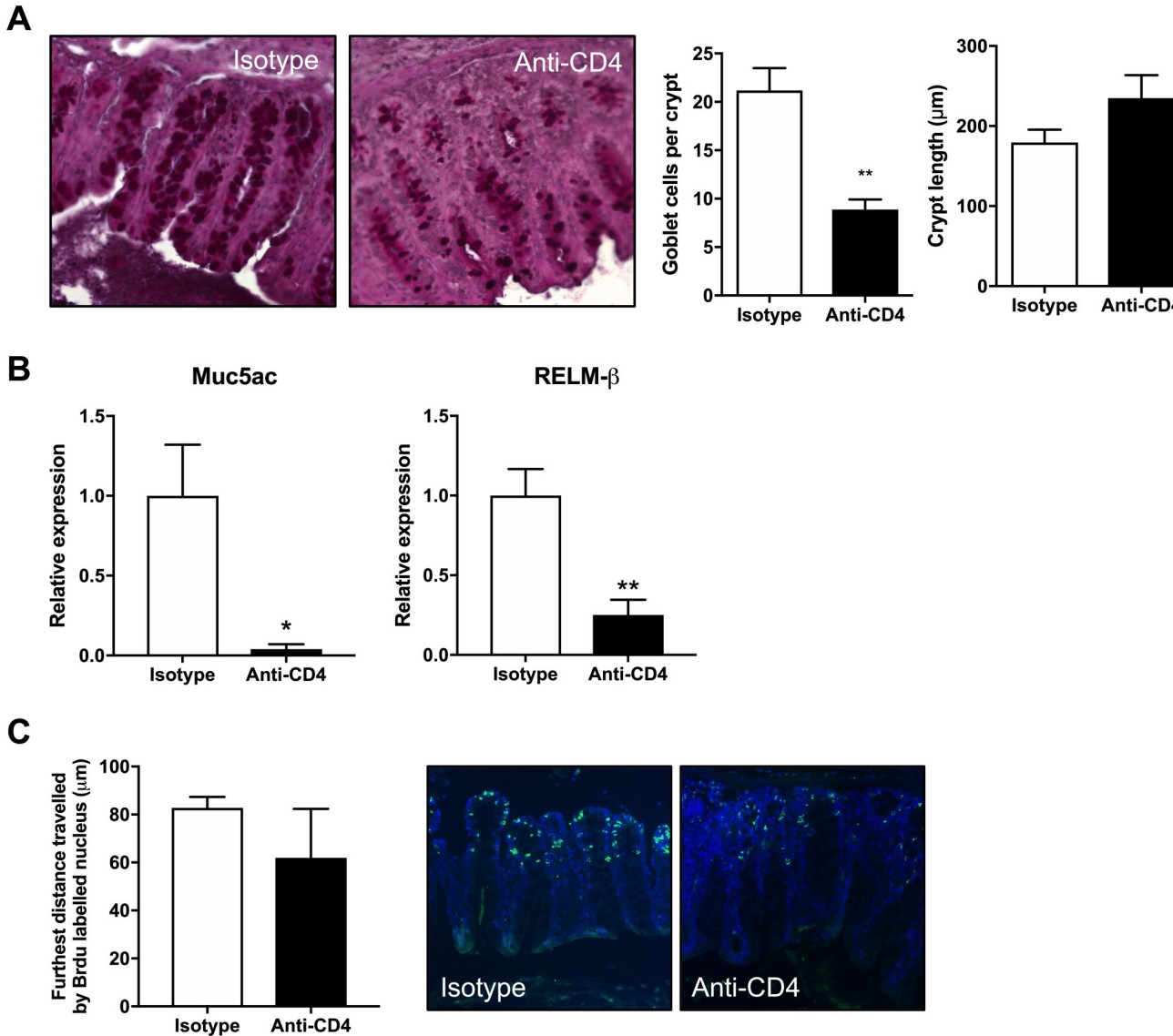

**Fig 6. Changes in worm expulsion mechanisms following CD4+ T cell depletion in trickle infection.** Mice were infected weekly for 9 weeks with 20 eggs and sacrificed at week 11 post primary infection. Between week 8 and week 10 CD4+ cells were depleted using neutralizing antibodies given 3 times a week for 3 weeks. Following CD4+ T cell depletion several parameters associated with worm expulsion were analyzed. (A) Caecal tissue was fixed in NBF at sacrifice and stained using PAS. Representative images of stained sections along with quantification of crypt length and goblet cell counts, given as the mean number per mouse quantified from a minimum of 10 crypts per mouse, n = 5. (B) Relative expression of Muc5ac and RELMβ analyzed by qPCR, n = 5. (C) Epithelial cell turnover was measured by I.P. injecting mice 9 hours prior to sacrifice with BrdU and quantifying the furthest BrdU+ve cell from the base of the intestinal crypt via microscopy. Sections stained by anti-BrdU (green) and DAPI (blue), n = 4–5. Statistics measured using an unpaired t test. Data presented as mean +/- SEM, * = p<0.05, ** = p<0.01.

In order to test whether CD4+ T cells from mice that had developed resistance following trickle infection could induce resistance to low dose infection, CD4+ T cells were purified from week 11 trickle infected mice and adoptively transferred into C57BL/6 mice and challenged with a low dose infection. Analysis of worm burdens on day 35 post infection showed no difference in worm burdens between cell recipients and control mice (S5C Fig). However, cell recipients did show slightly elevated levels of both parasite-specific IgG1 and IgG2c compared to controls (S5D Fig).

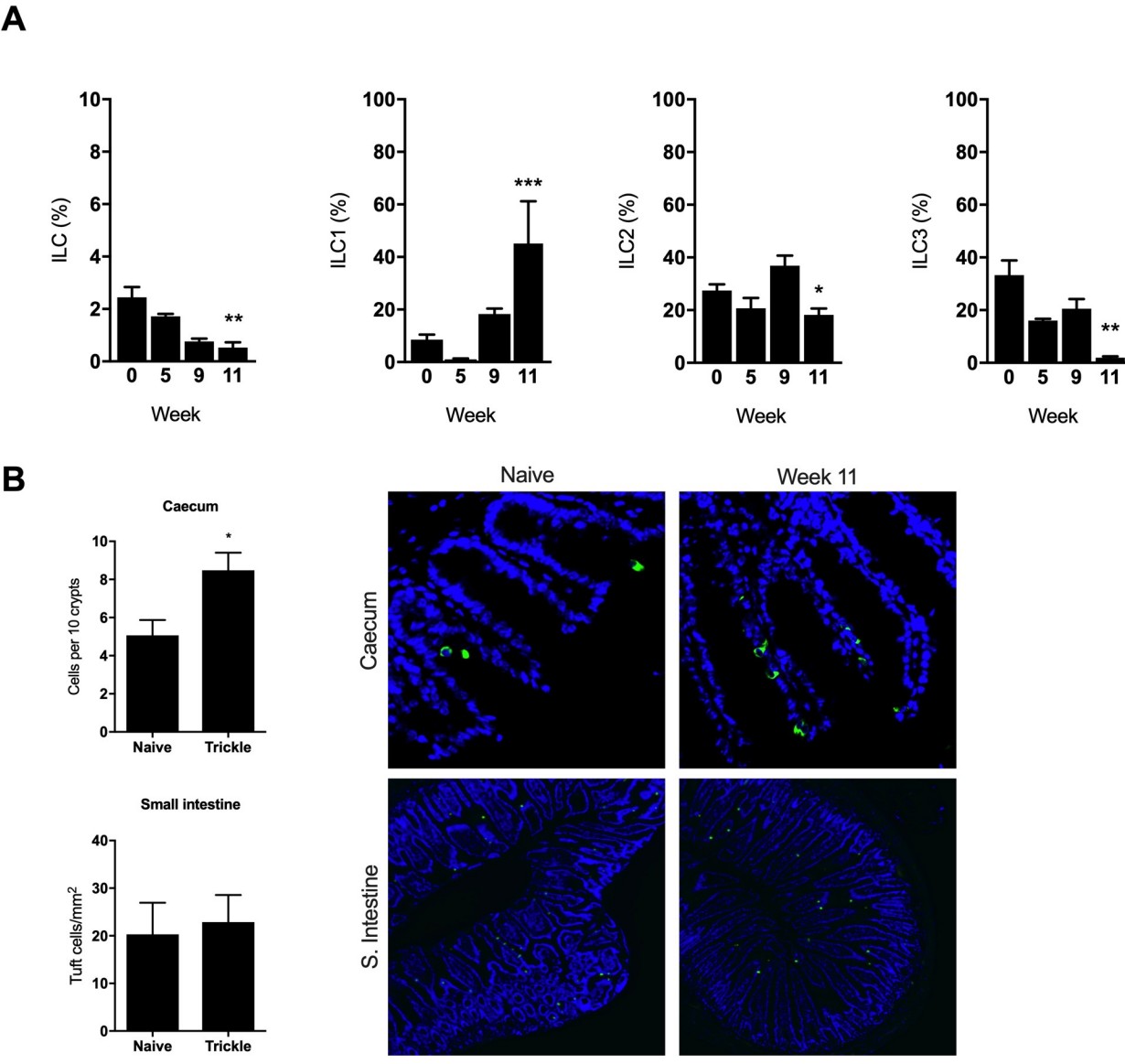

**Fig 7. Innate lymphoid cells and tuft cell proliferation in *T.muris* trickle infection.** (A) Proportion of innate lymphoid cells in the large intestine was quantified by FACS. ILCs were identified as lineage negative, CD90.2+CD127+ cells. Total ILCs are given as a percentage of live cells. Individual ILC subsets (ILC1, ILC2 & ILC3) are given as a percentage of total ILCs. (B) Tuft cells in caecal and small intestine sections of trickle infected mice quantified by microscopy. Tuft cells were identified as Dclk1 positive cells (green). DAPI stain in blue. n = 5, statistical analysis completed by a one way-ANOVA, * = p<0.05.

## Innate lymphoid cells play little role in resistance following *T. muris* trickle infection

ILCs have been shown to play indispensable roles during resistance in multiple model helminth infections including *N. brasiliensis* and *H. polygyrus* [28,29]. To determine the role of ILCs during *T. muris* trickle infection, ILCs were analysed by flow cytometry. Total intestinal ILC proportions decreased following the development of resistance at week 11 during trickle infection (Fig 7A). Total ILC numbers decreased over the infection period compared to naïve animals and this was reflected by a reduction in ILC2 and ILC3 proportions relative to ILC1s (Fig 7A). Analysis of ILC populations in the MLN showed no changes during the trickle

infection regime, including no changes in ILC2 populations (S6 Fig). Additionally, previous research has demonstrated that tuft cells proliferate following *N. brasiliensis* and *H. polygyrus* infection and play a key role in inducing expansion of ILC2s by secreting IL-25 [8]. During *T. muris* trickle infection there was a small but significant increase tuft cell numbers (identified by Dclk1 expression) in the caecum despite a significant decrease in ILC2s at this time point. No change in tuft cell numbers in the small intestine was observed (Fig 7B). To confirm the role of ILC2s during *T. muris* trickle infection, ICOS-T mice that can be specifically depleted for ILC2s by DTx treatment were used [29]. Mice were infected repeatedly using the week 11 trickle regime and received DTx or PBS control treatment at week 8–10 when resistance is observed to develop (Fig 8A). DTx treatment resulted in a significant reduction in total ILC2 in the caecum (Fig 8B). This depletion did not alter worm burdens and therefore the development of resistance following *T. muris* trickle infection (Fig 8C). High dose infections (~200 eggs in a single bolus) are a well-established model of Th2-dependent resistance to *T. muris* [30]. Using ICOS-T mice, we observed that ILC2 depletion following a single high dose infection did not alter worm expulsion relative to control mice (S7 Fig).

## Immune responses during trickle infection

A number of studies suggest that individuals naturally infected by helminths have altered susceptibility to atopy and asthma when assessed by skin prick testing to common allergens [31–33]. To determine whether *T. muris* trickle infection can alter the allergic response, *T. muris* infected mice were sensitized with the allergen OVA. Local immediate hypersensitivity in the skin was measured following OVA challenge and revealed that a neither a single low dose nor trickle *T. muris* infection altered the local immediate hypersensitivity response as compared to uninfected animals (S8 Fig).

In helminth endemic countries it is common to observe that individuals are infected with multiple helminth species including multiple GI nematodes such as whipworm and hookworm [34] and a variety of laboratory co-infection studies have shown that infection by one species of GI nematode can affect the response to another [35–37], although always with single high doses of the respective parasites. In order to investigate the effect of a trickle *T. muris* infection upon another GI nematode infection, mice were additionally challenged with a single high dose of *N. brasiliensis* a model of human hookworm infection. Co-infection following trickle (or indeed a single low dose infection by *T. muris*) did not alter worm expulsion kinetics of *T. muris* or *N. brasiliensis* (S9 Fig).

## Dysbiosis of the intestinal microbiome begins to resolve following the development of resistance in *T. muris* trickle infection

It has previously been demonstrated that chronic *T. muris* infection following a single low dose infection results in a dysbiotic microbiome [21]. Using 16S sequencing we compared changes in the composition of the final stool microbiota between low dose infections and trickle infection (S10 Fig). Both low dose and trickle infection induced shifts in the microbiota composition demonstrated by NMDS analysis (Fig 9A and 9B). Trickle infection promoted a more significant change in the biota at week 11 compared to the low dose infection. For low dose infection this change in the microbiota was largely consistent between week 9 and week 11. However, under a trickle infection, at week 11, a partial shift back towards the naïve groups was observed, coinciding with the development of resistance to infection. To look in more detail where these changes were occurring, alpha diversity of the most abundant bacterial phyla were quantified using Shannon indexes. Both infection regimes showed a significant reduction in overall Shannon diversity at week 9 (Fig 9C and 9D). However, at week 11 when resistance had developed, population

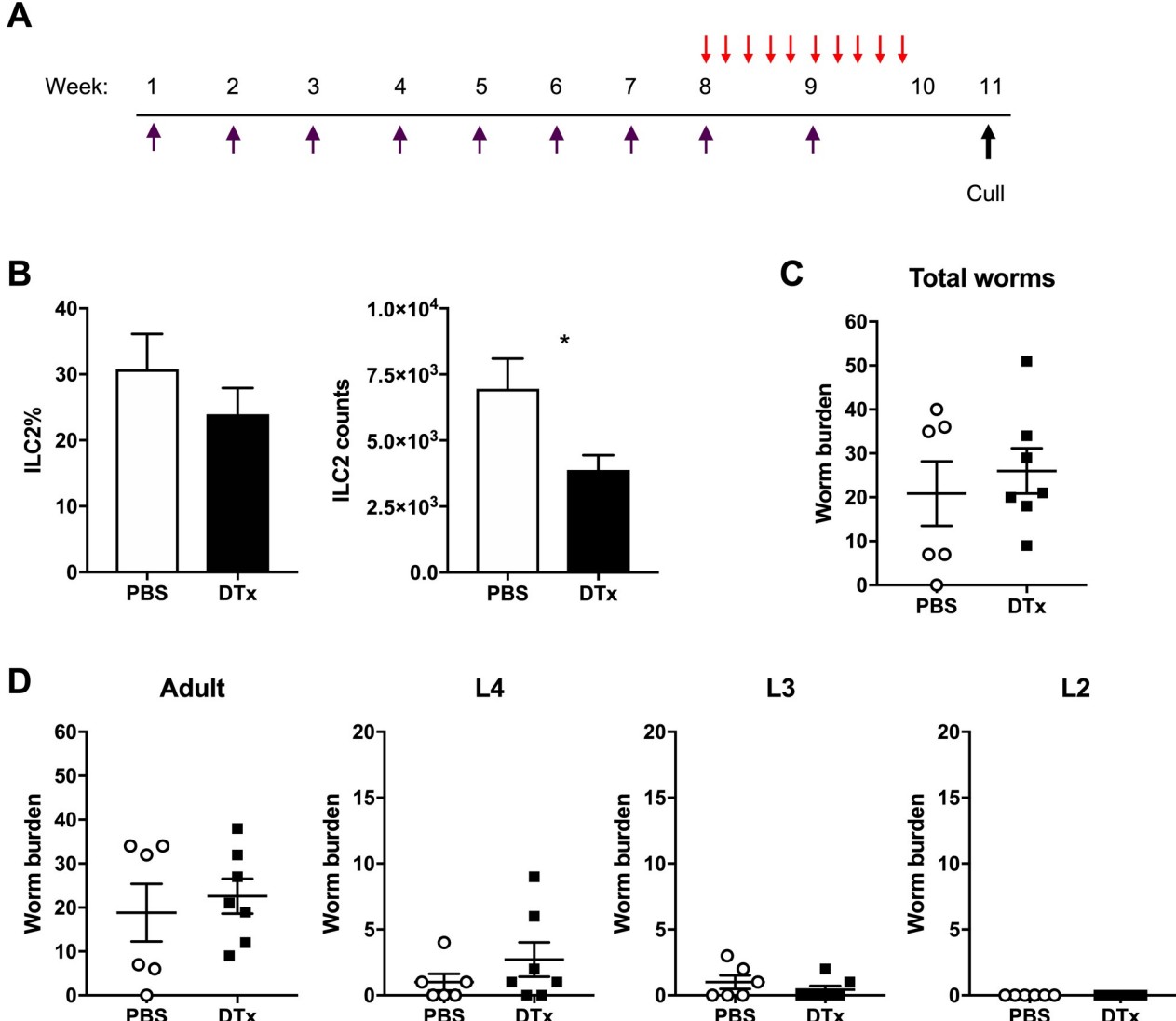

**Fig 8. Depletion of ILC2s in ICOS-T mice during *T. muris* trickle infection.** ILC2s were inducibly depleted from ICOS-T mice by DTx treatment. (A) Mice received 750ng DTx (red arrow) for 10 days at week 8–10 of trickle infection (purple arrow) where mice were infected weekly for 9 weeks with 20 eggs and culled at week 11 post primary infection. Control mice received PBS control injections at the same time points. Two weeks following the final trickle infection worm burdens and ILC2 depletion were analyzed. (B) Flow cytometry to confirm depletion, ILC2s identified as Lineage-, CD127+, CD90.2+, GATA3+ cells. Data are represented as a percentage of total ILCs and as total cell counts. (C) Total worm burdens of trickle infected mice. (D) Worm burdens of developmental stages during trickle infection. Statistical analysis completed by an unpaired t-test. Data presented as mean +/- SEM, * = p<0.05. n = 6–7 from two pooled experiments.

diversity began to recover in trickle infected mice (Fig 10D). This recovery of diversity appeared to predominate in the Bacteriodetes phylum (Fig 9D), and was not seen in low dose infection (Fig 9C). This trend could also be observed at the genus level, where some of the most abundant genera present in the naïve mice at both time points were diminished or could no longer be detected in trickle infected mice at week 9, but were present at week 11 (Fig 10).

## Discussion

Although our knowledge of the immune responses to gastrointestinal nematodes is continuously expanding, the majority of research follows infection after a single high or low dose of

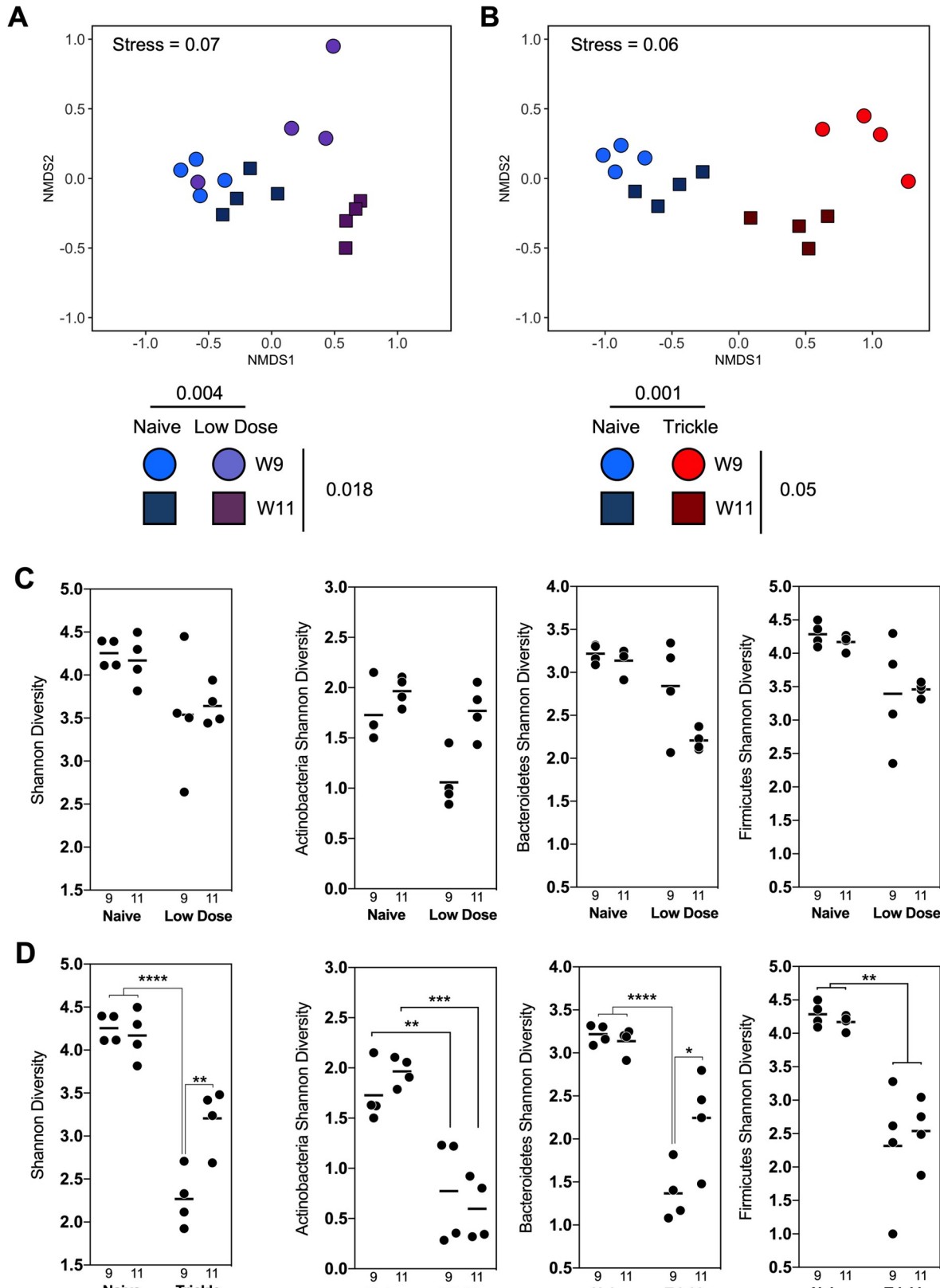

**Fig 9. NMDS analysis of fecal microbial communities during low dose and trickle *T. muris* infection.** (A) NMDS plot of microbial communities during *T. muris* low dose infection. Naïve mice (blue) were compared to mice trickled with *T. muris* (purple) at 9 and 11

weeks post infection (○ & □). Infection and time drove significant changes in microbiome composition (p = 0.001 & p = 0.05) assessed by PERMANOVA. (B) NMDS plot of microbial communities during *T. muris* trickle infection. Naïve mice (blue) were compared to mice trickled with *T. muris* (red) at 9 and 11 weeks post infection (○ & □). Infection drove a significant change in microbiome composition (p < 0.004) assessed by PERMANOVA. Axes represent a scale of Euclidian distances between samples where the centre is zero. Stress measures quality of fit (< 0.1 indicates a very good fit). (C & D) Alpha diversity of (C) low dose and (D) trickle infected mice given calculated in R by Shannon diversity test using the Vegan package. Significance is calculated by two-way ANOVA. Data presented as mean, * = p<0.05, ** = p<0.01, **** = p<0.0001.

eggs. However, *in situ* individuals experience repeated low doses of infection. As human studies are inherently limited and animal models following single infection often show conflicting results when compared to human studies, it is desirable to ensure we are correctly modeling natural infection. Here we functionally characterized the response to infection following a more representative regime by infecting mice repeatedly with low doses, named trickle infection, of the whipworm *T. muris*.

Following trickle infection, we show resistance developed concurrently with intestinal immune responses shifting from CD4+ Th1 cell dominated to a CD4+ Th2 cell dominated response. The development of Type 2 responses coincided with a significant reduction in both adult and larval worms. Absence of early larval stages suggests that the resistance was targeted/most effective against incoming larval stages. Adult worms, however, persisted until week 30 shown by continued low faecal egg output. Therefore, only partial resistance was induced, which is similar to that seen in naturally infected individuals with human whipworm [2]. Indeed, acquired immunity to most human STHs is slow to develop and incomplete, with

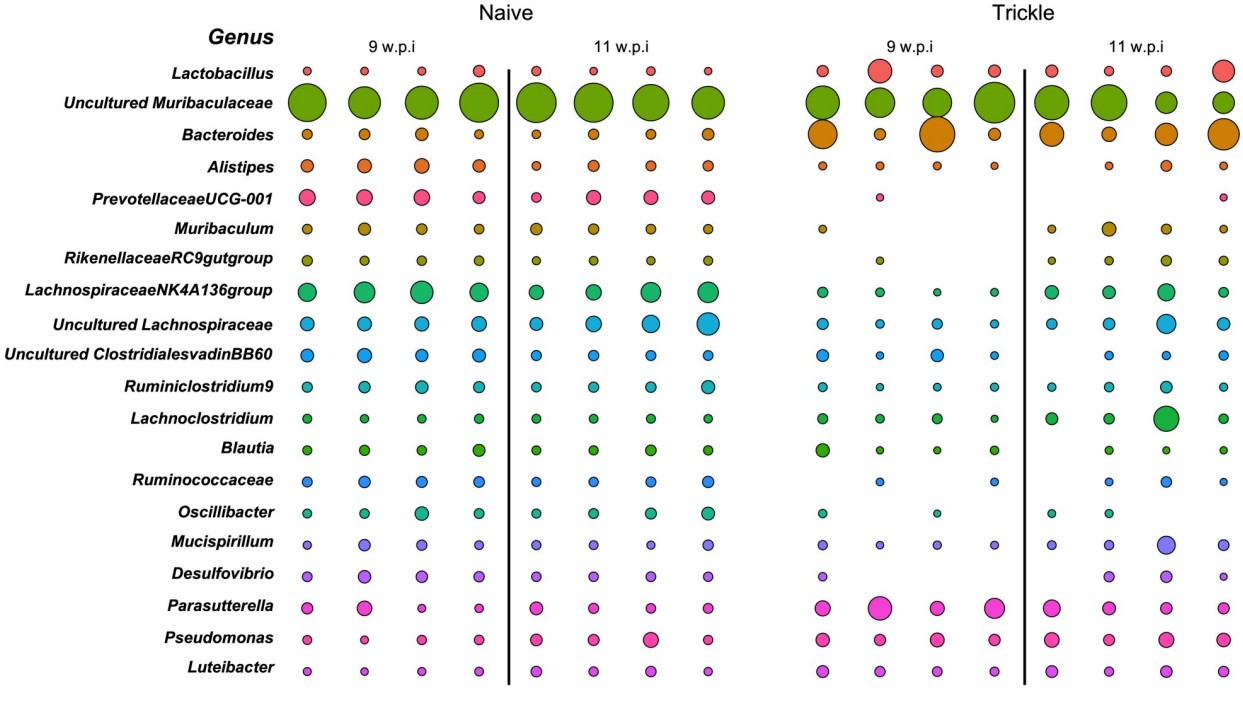

**Fig 10. Comparative abundance of bacterial genera in naïve and trickle infected mice.** Bubble plot representing the relative abundance of the top 20 genera determined as those with the highest median abundance across all individuals (as a proportion of total microbial composition). Circle size is representative of the proportion of the microbiota comprised by that genus. Empty spaces indicate that there was no detection of that genus by 16S sequencing.

'resistant' individuals in endemic regions harboring low numbers of parasites [2]. Previous studies using repeated *T. muris* low dose infection over a much shorter time frame (six infections over a 11 day infection period) suggested host genetics also influenced the capacity to generate resistance, although association between resistance and immune response type was difficult to define [38]. The acquisition of resistance following trickle infection with *T. muris* was associated with elevated IL-13 production and IL-13-mediated effector mechanisms including goblet cell hyperplasia, Muc5ac production, mucin sulphation, and increased epithelial cell turnover, as seen in previous studies using a single high dose infection to induce resistance [14,16,17,20]. Although immunity induced by trickle infection reduced worm numbers significantly it was not able to completely clear infection, as a low level of infection with adult worms persisted. The reasons for this are unclear although may be related to worm size and niche occupied by the parasite in combination with ongoing parasite induced immunomodulatory mechanisms [39]. For example, adult parasites are extensively embedded within epithelial tunnels, with the anterior of each worm "sewn" through thousands of epithelial cells [40,41] which physically presents a considerable challenge to remove. Previous work has suggested that the IL-13 controlled epithelial escalator was most successful at expelling worms at approximately day 14 post infection when smaller larval stages reside in the lower/mid region of the crypt where epithelial cells move faster [14]. Furthermore, following anti-helminthic treatment and challenge infection, trickle infected mice had reduced worm burdens but again were unable to completely expel worms, confirming only partial resistance is induced to even long term challenge. This would suggest that under conditions of repeated low dose infection such as encountered naturally, complete removal of the parasite burden may be difficult to achieve especially once adult worms are present and given that drug-treatment does not alter immune status significantly. It also predicts that infection would be highly prevalent in the population with most individuals infected by low numbers of worms.

The data presented here demonstrates a central role for CD4+ T cells in the development of resistance to *T. muris* trickle infection. When taken together with the adoptive transfer experiments the present data suggests that CD4+ T cells from trickle infected mice are necessary but may not be sufficient for resistance. However, the reasons for the lack of transfer immunity could be manifold e.g. non-optimum timepoint of harvest or transfer of cells relative to infection, too few cells etc. Also, as only partial immunity is evident following trickle infection, it is very likely that there are mixed populations of Th1/Th2 cells in the transferred population (supported by increases in both parasite-specific IgG1 and IgG2c) which may be less effective upon transfer. Thus, the adoptive transfer approach will require considerable further experimental investigation and dissection going forward. However, a critical role for CD4+ T cells is consistent with other models of resistance to *Tirchuris* infection [19,42]. Following CD4+ T cell depletion, worm expulsion and associated effector mechanisms were reduced. Robust *T. muris* specific antibody responses were generated during trickle infection, with high levels apparent prior to induction of resistance by week 11. Depletion of CD4+ T cells resulted in a mild reduction in parasite-specific IgG levels antibody unlikely to be responsible for diminished resistance to infection. Previous data from single high-dose infection studies in *T. muris* have demonstrated that antibody is not required for resistance [19]. In concert with this, mice deficient in FcγR are resistant to infection [43], as are mice deficient in *Aicda*, which are unable to class switch or develop high affinity antibody infection (3i consortium). Taken together, the dataset presented here suggests that antibody does not a play a major role in resistance induced by trickle infection, but it does not definitively show this, and further work is required to confirm this conclusion. Interestingly, this is not the case for other models of GI nematode infection such as *H. polygyrus* where antibody does play a role in host resistance particularly during a secondary infection [44]. This may reflect the different niches occupied by

the parasites underpinning the concept of multiple/redundant Type 2 controlled protective responses against these large multicellular pathogens. This is also evident in the present work regarding the role of ILC2 in host protection. IL-13 is key to resistance to multiple GI nematode infections and ILC2 are a potent source of this cytokine [28,29] with these cells sufficient for resistance to *N. brasiliensis* infection [29]. Here we suggest that ILC2s have little/no role in resistance against *T. muris* infection following either a trickle infection or a single high dose infection. A major driver of the ILC2 response following intestinal parasite infection has been identified as the tuft cell, which produces IL-25 promoting ILC2 expansion and IL-13 production [7,8,44]. The tuft cell response is muted during *T. muris* infection compared to other systems such as *N.brasiliensis*, *H. polygyrus* and *T. spiralis* [7,8,45]. Previous work has shown that IL-25 null mice are susceptible to a high single dose *T. muris* infection although no increase in IL-25 mRNA expression was found following infection in WT mice. Interestingly, resistance and a Th2 response was recovered in IL-25 null mice when IL-12 was blocked, suggesting that resistance did not absolutely require IL-25 [46]. The present data supports a poor IL-25 response following *T. muris* infection as evidenced by minimal changes in tuft cell numbers. The difference between *Trichuris* and other model systems studied may be due to both the infection regime/dose used and significant differences in life strategies evolved by the different species of parasite. Certainly, most models have used high dose bolus infections to stimulate strong Type 2 immunity including ILC2 responses. A strong induction of ILC2 and as a consequence, a considerable increase in IL-13, may provide rapid and sufficient induction of appropriate effector mechanisms required to expel the parasites [47]. Also, *T. muris* is distinct from other GI model systems in that to progress to patency it induces a Th1 response as part of its strategy to survive whereas even in systems where chronic infection occurs (e.g. *H. polygyrus* infection) it is set against induction and subsequent modulation of a Type2/Th2 response [48].

The intestinal region which *Trichuris* inhabits is also relatively unique when compared to other enteric helminths, with the caecum being its preferred site [16]. It is not surprising therefore that *T. muris* can induce an intestinal dysbiosis following chronic infection [21,22] that is dependent upon the presence of worms [21,23]. Here we used the final stool microbiota as proxy for the meta-state of the intestinal microbiome. The distal colon and caecum are anatomically distinct environments with variations in histological structure and thickness of the mucus barrier, meaning that there will be some distinction between the composition of the microbiota between sites. However, we have previously shown, using low dose infection, that stool microbiota correlates closely with caecal content during *T. muris* infection [21]. During trickle infection we now show a dysbiosis in the intestinal microbiome, distinct from single low dose infected mice, that begins to resolve once partial resistance develops. We found that high abundance genera present in naïve mice that were lost or reduced at 9 weeks p.i. in trickle infected mice were found to return following the observed development of resistance, as well as an overall increase in alpha diversity. It is tempting to speculate that given sufficient time the microbiota would return to a naïve-like phenotype despite the continued presence of *T. muris*. Microbiome studies from humans naturally infected with GI nematodes including whipworm have been carried out with varied outcomes regarding the influence of helminth infection [49–51]. This may be explained in part by cofounding factors including co-infection, diet, age and level of worm burden. However, it has been suggested that *T. trichiura* infection alone is not sufficient to alter the intestinal microbiome [50]. Our data supports a more dynamic view of the relationship between the microbiota, and infection and immunity with *Trichuris* species. We suggest that during early recurrent infections, when the host is still susceptible, *Trichuris* alone is capable of driving dysbiosis. However, at the point where a protective immunity is attained, the microbiota will begin to return to a more homeostatic composition characteristic of un-infected individuals.

There is conflicting data as to the influence of chronic helminth/GI nematode infection upon the incidence of allergic responses e.g. skin prick tests in humans [52]. With regards to model systems studies using chronic *H. polygyrus* infections have shown that allergic lung responses are muted compared to non-infected animals [53,54] and single low dose chronic infections with *T. muris* also influence lung allergic responsiveness [55]. Low dose chronic *T. muris* infection also mutes contact hypersensitivity response to Th1 contact sensitizing agents but not Th2 sensitizing agents [16]. Chronic *H. polygyrus* infection modulated responses to both Th1 and Th2 skin sensitizing agents [16]. Here, we show that trickle *T. muris* infection does not modulate an immediate (IgE) hypersensitivity response in the skin (nor does a single low dose infection).

It is common in endemic regions for individuals to be infected by multiple species of GI nematode, including skin penetrating species e.g. hookworm [56–58]. In order to assess whether such an infection would influence or be influenced by a *T. muris* trickle infection a single dose of *N. brasiliensis* was superimposed. Neither parasite appeared to be affected in terms of resistance status at least with regards to the timing and challenge regime used here.

Overall, the present work demonstrates that partial Type 2 mediated resistance to whipworm, similar to that seen under natural infection, can be generated under laboratory conditions using repeated low dose infection. Functionally, immunity is dependent upon CD4+ Th2 cells and a role for ILC2s could not be established. This approach will complement and extend those models already used for study of immunity to GI nematodes and importantly highlight key differences allowing a more rational comparison to the field situation.

## Materials and methods

### Ethics statement

Experiments were performed under the regulation of the Home Office Scientific Procedures Act (1986), and were approved by the University of Manchester Animal Welfare and Ethical Review Body.

### Animals

C57BL/6, RAG-/- and iCOS-T (kindly provided by Dr A N McKenzie),[29] mice were maintained in individually ventilated cages in the University of Manchester animal facility in accordance with Home office regulations (1986). Mice were housed for at least 7 days prior to experimentation and were 6–8 weeks old males.

### Parasitological techniques

Mice were infected with *Trichuris muris* embryonated eggs weekly for 3, 5 or 9 weeks. Mice received 20 eggs in 200μl of $dH_20$ by oral gavage. Two weeks following the final infection, worm burdens and immune responses were analysed. Caeca were collected from infected mice and parasites sorted into L2, L3, L4 larval stages and adult worms for counting. Faecal egg output was analysed by suspending stool pellets in saturated NaCl and counting in a McMaster Chamber, eggs per g of faeces was calculated.

Mice were infected with 500 *Nippostrongylus brasiliensis* stage 3 larvae (L3) by subcutaneous injection. To determine worm burden of *N. brasiliensis* the lungs and small intestine of infected mice were collected at autopsy. The lungs were placed onto gauze and chopped into small pieces. The gauze containing lung tissue was suspended in 50ml of PBS and incubated at 37˚C for 4 hours to allow parasites to move from the tissue and to be counted. The lung tissue was then digested in 1mg/ml collagenase D (Sigma) 0.5mg/ml Dispase (Sigma) in RPMI 1640 medium and incubated at 37˚C on a rotator. Digested lung tissue was centrifuged and re-

suspended in PBS and parasites were counted. Small intestine was split longitudinally and place in gauze suspended in PBS. After incubation at 37˚C for 4 hours, worms were counted.

## Production of *T. muris* Excretory/ Secretory product

To produce *T. muris* excretory/secretory products (E/S) for immunological techniques, the parasite was passaged through SCID mice that are susceptible to infection. SCIDs received a high dose of approximately 400 *T. muris* embryonated eggs and at approximately day 35–42 p.i. the large intestine was collected to produce adult E/S. To produce L4 E/S mice were culled at day 28 p.i. To produce L3 E/S mice were culled at day 20 p.i. Guts were collected and longitudinally split open and washed in warmed 5x penicillin/streptomycin in RMPI 1640 medium. Adult, L4 and L3 worms were carefully pulled from the gut using fine forceps and transferred to a 6 well plate containing 4ml warmed 5x pen/strep in RMPI 1640 medium. Plates were placed into a moist humidity box and incubated for 4 hours a 37˚C. Adult worms were then split into 2 wells containing fresh medium and incubated again in a humidity box at 37˚C overnight. Supernatant from 4 hour and 24 hour incubation was collected and centrifuged at 2000g for 15 minutes. *T. muris* eggs from adult worms were resuspended in 40ml deionised water and filtered through a 100μm nylon sieve before transferring to a cell culture flask. To allow embryonation, eggs were stored in darkness for approximately 8 weeks and then stored at 4˚C. Susceptible mice were subsequently infected with a high dose infection to determine infectivity of each new batch of eggs.

To produce L2 E/S, SCID mice were infected with a high dose of *T. muris* and at day 14 p.i. guts were collected and placed in 5 x pen/strep in PBS. Guts were cut longitudinally and were washed to remove faecal debris. The guts were cut into small sections and added to 0.9% NaCl in PBS and incubated in a water bath at 37˚C for two hours to allow L2 larvae to come free from the epithelium.

L2 larvae were removed from the NaCl and placed into 5 x pen/strep RPMI 1640 medium and incubated overnight at 37˚C. The following day the larvae and RPMI 1640 medium was centrifuged at 720g and the E/S was recovered.

To produce L1 E/S, eggs were hatched in 2ml sodium hypochlorite (Fisher chemical) in 4ml $H_2O$ for 2 hours at 37˚C. L1 larvae were washed in RPMI 1640 medium, 10% FCS, 100 unit/ml of penicillin, 100μl/ml of streptomycin until media returned to original colour. Larvae were cultured at 37˚C for 3 weeks with media being collected and replaced twice a week.

All E/S supernatant was collected and filter sterilised through a 0.2 μm syringe filter (Merck). E/S was concentrated using an Amicon Ultra-15 centrifugal filter unit (Millipore) by spinning at 3000g for 15 minutes at 4˚C. E/S was dialysed against PBS using Slide-A-Lyzer Dialysis Cassettes, 3.500 MWCO (Thermo Science) at 4˚C. The concentration of E/S was measured using the Nanodrop 1000 spectrophotometer (Thermo Fisher Science) and aliquoted before storing at -20˚C.

## CD4+ T cell depletion, adoptive transfer, and treatment of iCOS-T mice with diphtheria toxin for innate lymphoid cell 2 depletion

*In vivo* depletion of CD4+ T cells was achieved by administration of rat IgG2b anti-mouse CD4 (GK1.5, BioXCell). Control animals were treated with the matched isotype control antibody (LTF-2, BioXCell). Animals were treated 3 times a week for 3 weeks with 200μg antibody in 200μl by intraperitoneal (IP) injection.

ICOS-T mice were treated with diphtheria toxin (DTx, Merck) to induce specific depletion of ILC2s. Mice received 750ng DTX in 200μl by IP injection once a day for 5 days. Control mice received a PBS control injection (20).

To isolate CD4+ T cells, single MLN cell suspensions were prepared and purified using (L3T4) MicoBeads (Miltenyi Biotec). Purified CD4+ T cells were counted and resuspended in 0.1% BSA/PBS for injection.

## CD4+ T cell and innate lymphoid cell quantification by Flow cytometry

Lymph nodes were pressed through a 100μm nylon cell strainer (Fisher Scientific) and cells were pelleted by centrifugation at 400g for 5 minutes. The supernatant was removed and the pelleted MLN cells were resuspended in 1ml of complete RPMI.

The large intestine was opened longitudinally with blunt ended scissors and cut into 0.5cm segments before washing in 2% foetal calf serum (FCS) in Hank's Balanced Salt Solution (HBSS) (Sigma, Life Sciences). To remove epithelial cells, large intestine segments were added to 2mM Ethylenediaminetetraacetic acid (EDTA)/HBSS and incubated on a rotator for 15 minutes at 37˚C. Samples were strained through a metal strainer and were washed in 2% FCS HBSS. Gut segments were incubated in EDTA/HBSS for 30 minutes at 37˚C on a rotator. To isolate lamina propria lymphocytes (LPLs), large intestine segments were added to an enzyme cocktail of 0.85mg/ml Collagenase V (Sigma), 1.25mg/ml Collagenase D (Roche), 1mg/ml Dispase (Gibco, Life Technologies) and 30μg/ml DNase (Roche) in 10ml complete RPMI (Sigma, Life Sciences). Samples were incubated on a rotator at 37˚C for 45 minutes or until all tissue was digested. Samples were passed through a cell strainer (Fisher Scientific resuspended in complete RPMI.

MLN and large intestine cell suspension were stained with 1μl of Fixable Viability Dye (eBioscience) to identify live cells. Samples were fixed and permeabilised using the Foxp3/ Transcription Factor Staining Buffer Set (eBioscience) and blocked for non-specific binding by incubating samples in 50μl Anti-Mouse CD16/CD32 Fc Block (eBioscience). Samples were stained for cell surface and intracellular markers. Samples were read on a BD LSRFortessa flow cytometer (BD Biosciences) running FACSDiva acquisition and analysed using FlowJo X (Tree Star, Inc).

Cell surface markers: primary antibodies anti-CD11b (CR3a); anti-NK1.1 (PK126); anti-Ly6G (Gr-1); anti-CD11c (N418); anti-CD45R (RA3-6B2); anti-CD19 (1D3); anti-TER-119 (TER-119); anti-CD49b (DX5); anti-F4/80 (BM8); anti-FCεR1 (MAR1); anti-CD4 (RM4-5); anti-CD45 (30-F11); anti-CD127 (A7R34); anti-TCR-beta (H57-597) and anti-CD8a (53–6.7) purchased from eBioscience. Primary anti-CD90.2 (30-H12) and secondary streptavidin PE Dazzel purchased from Biolegend. Intracellular markers: anti-GATA3 (16E10A23); anti-Tbet (ebio4BIO); anti-Roryt (B2D); anti-FOXP3 (FJK-16s) purchased from ebioscience. Primary antibodies were used at dilutions 1:50–1:800. Secondary antibodies used at 1:1000 dilution.

## Quantification of parasite specific IgG1 and IgG2a/c by ELISA

Blood was collected from mice at autopsy and serum was isolated. ImmunoGrade plates (BrandTech Scientific, Inc) were coated with 5μg/ml *T. muris* excretory secretory (E/S) product from Adult, L4, L2, L2 or L1 parasites. Plates were washed using a Skatron Scan Washer 500 (Molecular Devices, Norway) 5 times with PBST following each incubation. Plates were blocked with 3% bovine serum albumin (BSA). A double dilution of serum samples was added to plates with a starting dilution of 1:20. Biotinylated rat anti-mouse IgG1 (1:2000, Biorad) or rat anti-mouse IgG2a/c (1:1000, BD Pharmigen) was added to plates followed by Streptavidin peroxidase (Sigma). Plates were developed with ABTS (10% 2,2'azino 3-thyl benzthiazoline) and read at 405nm with 490nm reference on a Dynex MRX11 microplate reader (Dynex Technologies).

## Histology and immunofluorescence

Caecal tips were collected and fixed in Carnoy's solution. Sections were paraffin embedded and 5μm sections were mounted onto slides for staining. To analyse caecal crypts and goblet cell counts sections were stained with periodic acid Schiff's reagent (PAS) and counterstained with Meyers haematoxylin(Sigma). To analyse mucin sulphation, sections were stained with High- Iron Diamine-Alcian Blue (HID-AB). Images were visualised using an Axioskop upright microscope using the Axiovision software.

To stain for Brdu, Muc2 and tuft cells, sections were stained with primary anti-BRDU (BU1/75, BioRad), anti-Muc2 antibody, 1:200 dilution (5501, gift from D. Thornton) or anti-Dclk1 antibody, 1:800 dilution (Abcam) at 1:800 dilution). Sections were washed in PBS and incubated in secondary antibody goat anti-rabbit Af488 (Life techologies) for 1 hour at room temperature. Nuclear structures were stained with DAPI. An Olympus BX51 upright microscope was used to visualise staining using MetaVue software.

## Intestinal microbiome analysis

Stool samples were collected from mice and DNA was extracted using the QIAamp DNA stool mini kit (Qiagen) according to manufacturer's instructions. To charactarise the gut microbiome the 16S rRNA gene was amplified using 16S primers that target the V3/V4 region. Forward primer: 5'-TCGTCGGCAGCGTCAGATGTGTATAAGAGACAGCCTACGGGNGGC WGCAG-3'. Reverse primer: 5'-GTCTCGTGGGCTCGGAGATGTGTATAAGAGACAGGA CTACHVGGGTATCTAATCC-3'. Samples were amplified using the following conditions: 10–50 ng of template DNA, 12.5 μL of 2x KAPA HiFi HotStart ReadyMix, 5 μL each of 1 μM forward and reverse primers in a 25 μL reaction. The cycle parameters are as follows: 95˚C 3 min, then 25 cycles of 95˚C 30 s; 55 $^o$C 30: 72 $^o$C s: with a final step of 72 $^o$C for 3 min. The 16S amplicons were purified using AMPure XP magnetic bead purification. Samples were indexed using the Nextera XT index kit and then quantified by Illumina Miseq in the Genomic Technologies Core Facility at the University of Manchester. Sequences were trimmed using Trimmomatic then clustered into OTUs with a sequence similarity of 97% and taxonomy was assigned in QIIME [59] using the SILVA132 database. Statistical analysis and non-metric Multidimensional Scaling (NMDS) were performed using the R-package. OTU counts were normalised using the DESeq2 package [60]. Shannon diversity, PERMANOVA tests, and rarefaction were performed using the Vegan package. NMDS plots represent Euclidian distances plotted in arbitrary two dimensional space centred on zero. Stress is a measure of quality of fit of Bray-Curtis dissimilarities where less than 0.2 indicates good fit on two dimensional plots. NMDS and bubble graphs were produced using the ggplot2 package.

## Gene expression of gut associated genes

Caecal tips were collected at autopsy and RNA was extracted using the TRIzol method (Invitrogen). cDNA was generating using the GoScript Reverse Transcriptase kit (Promega). Quantitative-PCR was set up using the SensiFAST SYBR Hi-ROX kit (Bioline) on a StepONE system (Applied Biosytems). Genes of interest were normalised against β-actin and expressed as fold change compared to naïve mRNA expression. Muc2: F:5'GTCCAGGGTCTGGATCA CA. R:5' CAGATGGCAGTGAGCTGAGC. Muc5ac: F:5' GTGATGCACCCATGATCTATT TTG R:5; ACTCGGAGCTATAACAGGTCATGTC. Relm-β: F:5' GCTCTTCCCTTTCCTT CTCCAA R:5' AACACAGTGTAGGCTTCATGCTGTA. TFF3: F:5' TTATGCTGTTGGTG GTCCTG. R:5' CAGCCACGGTTGTTACACTG. β-actin: F: 5' TCTTGGGTATGGAATGT GGCA R:5'ACAGCACTGTGTTGGCATAGAGGT. TSLP: F:5' AGCAAGCCAGTCTGTCTC GTGAAA R: TGTGCCAATTCCTGAGTACCGTCA Amphiregulin: F:5' TCTGCCATCATC

CTCGCAGCTATT R:5' CGGTGTGGCTTGGCAATGATTCAA. Bacterial load: F: 5'
TCCTACGGGAGGCAGCAG. R: 5' GGACTACCAGGGTATCTAATCTT.

## Mucus extraction and analysis

Mucus was extracted by flushing large intestines with PBS and shaking in 2M urea. Samples
were reduced by dithiothreitol (DTT) and run on a 1% agarose gel in TAE buffer. Mucins
were transferred onto a nitrocellulose membrane, blocked with casein and probed using
chicken anti-mouse Muc5ac antibody (Rockland) followed by an incubation with the goat
anti-chicken IgY AF790 (Abcam). The membrane was imaged using the Odyssey CLx Imaging
system (Licor) on Image studio software (15).

## MLN re-stimulation and cytokine analysis by Cytometric bead assay (CBA)

Mesenteric lymph nodes (MLNs) were isolated at autopsy and cells were restimulated with
50µg/ml *T. muris* adult ES. Cells were incubated for 48 hours at 37˚C, 5% $CO_2$. Supernatant
from cell cultures were incubated with a capture bead cocktail for cytokines of interest (BD
Bioscience), containing one for each cytokine. Detection beads (BD Bioscience), diluted in
detection reagent (BD Bioscience) was added to each well and incubated. Plates were washed
and resuspended in 70µl wash buffer (BD Bioscience). Cytokines were measured on a MACS-
Quant Analyser (Miltenyi Biotec) and analysed using the FCAP array software in reference to
a standard curve.

## Measurement of skin immediate hypersensitivity

Mice were sensitised with 50µg ovalbumin (OVA, Sigma) in 2mg Alum (Thermo Scientific) by
I.P. injection. Control mice were injected with PBS in Alum. Mice received one dose a week
for 3 weeks. Two weeks after the final injection the anaphylaxis assay was performed.

Mice were anesthetised by 2% isoflurane. Mice were injected subcutaneously with 5µg
OVA in 10µl PBS into one ear and 10µl PBS into the other. The ear was stabilised onto a falcon
tube to aid injections. After 3 minutes 200µl of 0.5% Evans Blue dye (Sigma) was injected into
the tail vein. After 10 minutes, mice were euthanised and ears were removed and placed into
700µl Formamide (Sigma). The ears were incubated overnight at 63˚C to allow dye to leak into
the Formamide. 300µl of each sample was transferred into a 96 well plate in duplicate and read
on a VersaMax Microplate reader (Molecular Devices) at 620nm.

## Statistical analysis

Statistical analysis completed using a one-way ANOVA, followed by post-hoc Tukey's test or
an unpaired t-test using GraphPad Prism 7 software.

## Supporting information

**S1 Fig. *T. muris* trickle infection regime.** Schematic representing a trickle infection regime.
C57BL/6 mice were infected weekly with low doses of *T. muris* (20 eggs) for 3, 5, 7 or 9 weeks
(red arrow). 2 weeks after the final infectious dose at week 5, 7, 9 and 11, worm burdens and
immune responses were analysed (black box).
(TIF)

**S2 Fig. CD4+ T cells in MLN following *T. muris* trickle infection.** C57BL/6 mice were
infected weekly with low doses of *T. muris* and 2 weeks after the final trickle infection CD4+ T
cells in the MLN were analyzed by FACs. CD4+ cell subsets were identified by transcription
factor expression; Th1 (Tbet+), Th2 (GATA3+), Th17 (RORγt+), Tregs (FOXP3+). n = 10,

from two independent experiments. CD4+ T cell percentage calculated as percentage of all live cells. T cell subset percentage calculated as percentage of all CD4+ cells.
(TIF)

**S3 Fig. *T. muris* specific antibody responses in trickle infected mice.** Antibody responses measured from sera collected from trickle infected mice. Antibody levels were measured using indirect ELISA where serially diluted serum from individual mice was incubated in 96-well plates coated with *T. muris* E/S, then targeted with antibodies against mouse IgG1 or IgG2a/c. Values are given as arbitrary optical density values of the substrate measured at 405 nm. The antibody response specific for adult worms and larval stages 1–4 was measured. (A) IgG1 response. (B) IgG2a/c response. (C) Total IgE response. n = 5, statistical analysis completed by a one way-ANOVA. Data presented as mean +/- SEM, * = p<0.05, ** = p<0.01, **** = p< 0.0001.
(TIF)

**S4 Fig. Challenge infection of *T. muris* trickled mice.** To determine whether trickle infection could protect against a challenge infection, trickle infected mice were either left to expel all worms naturally or worms were removed by anti-helminthic treatment. (A) At week 30, following either a single high or low dose infection or trickle of low dose infections, when no worms were present, determined by measuring faecal egg output, mice were challenged with a single low dose infection. Control mice received a low dose challenge at week 30 post infection n = 10 or greater. (B) Following a single low dose infection or a trickle of 3 low dose infections, mice were treated with anti-helminthics to remove final worms at week 11 post infection. Worm expulsion was confirmed by the absence of eggs in the faeces, Mice were then challenged with a low dose infection one week following anti-helminthic treatment. Worm burden was assessed by eye under dissecting microscope. n = 5 representative of two independent experiments, statistical analysis completed by a one way ANOVA or an unpaired t test. Data presented as mean +/- SEM, * = p<0.05, ** = p<0.01, *** = p<0.001, **** = p< 0.0001.
(TIF)

**S5 Fig. *T. muris* specific antibody responses following CD4+ T cell depletion.** Sera from *T. muris* infected mice depleted of CD4+ T cells was collected and IgG1 and IgG2a responses specific for *T. muris* larval stages were quantified. Antibody levels were measured using indirect ELISA where serially diluted serum from individual mice was incubated in 96-well plates coated with *T. muris* E/S, then targeted with antibodies for against mouse IgG1 or IgG2a/c. Values are given as arbitrary optical density values of the substrate measured at 405 nm. A) IgG1 response to adult worms and larval stages 1–4. B) IgG2a response to adult worms and larval stages 1–4. Isotype control in grey. Anti-CD4 treatment mice in black n = 5. (C-D) CD4+ T cells were isolated and purified from week 11 trickled infected mice. $2x10^6$ CD4+ T cells were injected i.v. into C57BL/6 mice which then received a single low dose infection (20 eggs) the following day. C) Worm burden was counted at day 35 p.i. D) ELISA to quantify *T. muris* specific IgG1 and IgG2c levels. Data presented as mean +/- SEM, n = 5.
(TIF)

**S6 Fig. ILCs counts in MLN.** Innate lymphoid cell counts and percentage in the MLN following *T. muris* trickle infection measured by FACS, identified as lineage negative, CD90.2+, CD127+. Total ILC percentage calculated as percentage of all live cells. ILC subset calculated as the percentage of total ILCs. n = 3, statistical analysis completed by a one way-ANOVA. Data presented as mean +/- SEM, * = p<0.05, ** = p<0.01
(TIF)

**S7 Fig. Depletion of ILC2s in ICOS-T mice.** ILC2s were depleted from ICOS-T mice by DTx treatment. (A) Mice received 750ng DTx (red arrow) for 5 days before *T. muris* high dose (400 eggs) infection (purple arrow) and 1 week after *T. muris* infection for 5 days. Control mice received PBS control injections at the same time points. Worm burden was analysed at week 5 p.i. (black arrow). (B) Flow cytometry to confirm depletion, ILC2s identified as Lineage-, CD127+, CD90.2+, GATA3+ cells. ILC2% of all ILCs and ILC2 counts. (C) Worm burden of *T. muris* at day 35 p.i. Statistical analysis completed by an unpaired t-test. Data presented as mean +/- SEM, * = p<0.05. n = 3–5.
(TIF)

**S8 Fig. Immediate hypersensitivity response to OVA antigen during *Trichuris muris* infection.** C57BL/6 mice were infected with a single low dose or trickled with *T. muris* over 9 weeks. At week 8 following the first infection mice were sensitized with 50 μg OVA antigen in 2mg of Alum or PBS control for 3 weeks. 4 weeks following the final sensitization all mice were challenged with 50 μg OVA by intradermal injection in the right ear and a PBS control injection in the left ear. A) Timeline of *T. muris* infection and sensitisation and challenge of OVA/PBS. B) Immediate hypersensitivity response in mice following PBS and OVA challenge. Hypersensitivity was determined by measuring skin permeability after OVA challenge using an Evans Blue assay. Levels of Evans Blue were quantified by absorbance and data are represented as arbitrary absorbance values at 602 nm. C) Side-by-side comparison of hypersensitivity response following OVA stimulation in naïve mice, low dose infected mice, and trickle infected mice from panel B. D) Worm burdens of *T. muris* trickle infected mice. Data presented as mean +/- SEM. Statistical test calculated by an unpaired t-test, ** = p<0.01, *** = p<0.001, **** = p< 0.0001, n = 5.
(TIF)

**S9 Fig. Co-infection of *Trichuris muris* with *Nippostrongylus brasiliensis*.** C57BL/6 mice were infected with a single *T. muris* low dose (20 eggs) or trickle infected of 9 weekly low doses by oral gavage. At week 10, *T. muris* infected mice and naive mice were infected with a single high dose (300 larvae) of *N. brasiliensis* by subcutaneous injection. At day 3 and day 6 following the *N. brasiliensis* infection, worm burdens of *N. brasiliensis* from the lung and small intestine and *T. muris* in the caecum, were counted. A) Timeline of infection regime. B) Worm burdens of *N. brasiliensis* in the small intestine and lung. C) *T. muris* worm burdens from low dose infected and trickle infected mice. Burdens of total worms and well as adult, L4, L3 and L2 were counted for trickle infected mice. Data presented as mean +/- SEM. Statistical analysis was carried out using a one-way ANOVA followed by post-hoc Tukey's test. n = 5.
(TIF)

**S10 Fig. Composition of microbial communities during infection.** Following trickle infection mice were sacrificed at week 9 and week 11 post primary infection and the final stool microbiome was measured using 16S sequencing. (A) Rarefaction curves for individual samples calculated in R using the vegan package. (B) Phylum level comparisons between groups. Data represents the mean from 4 mice. Phyla representing, on average, less than 2% of the population were grouped into "other."
(TIF)

## Acknowledgments

We thank A. McKenzie (LMB) for providing iCOS-T mice; A. Bancroft for help with infection; L. Campbell for help with immediate hypersensitivity experiments; M. Lawson for advice and

feedback on microbiota analysis. We also thank the Faculty of Biology, Medicine & Health core facility services at the University of Manchester, including Genomic Technologies, Histology, Flow Cytometry, BSF, and Bioimaging. In particular we thank P. Wang in the Bioinformatics Core Facility for his help in the processing of sequencing data.

## Author Contributions

**Conceptualization:** David J. Thornton, Richard K. Grencis.

**Data curation:** Maya Glover, Stefano A. P. Colombo.

**Formal analysis:** Maya Glover, Stefano A. P. Colombo.

**Funding acquisition:** David J. Thornton, Richard K. Grencis.

**Investigation:** Maya Glover, Stefano A. P. Colombo.

**Methodology:** Maya Glover, Stefano A. P. Colombo.

**Project administration:** Maya Glover, Stefano A. P. Colombo, Richard K. Grencis.

**Resources:** Richard K. Grencis.

**Supervision:** David J. Thornton, Richard K. Grencis.

**Writing – original draft:** Maya Glover, Stefano A. P. Colombo, Richard K. Grencis.

**Writing – review & editing:** Stefano A. P. Colombo, Richard K. Grencis.

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
