## [Decision Letter · Decision Letter 0]

26 Jul 2019

Dear Prof. Grencis,

Thank you very much for submitting your manuscript "Trickle infection and immunity to Trichuris muris" (PPATHOGENS-D-19-01054) for review by PLOS Pathogens. Your manuscript was fully evaluated at the editorial level and by independent peer reviewers. The reviewers appreciated the attention to an important problem, but raised some substantial concerns about the manuscript as it currently stands. These issues must be addressed before we would be willing to consider a revised version of your study. We cannot, of course, promise publication at that time.

We therefore ask you to modify the manuscript according to the review recommendations before we can consider your manuscript for acceptance. Your revisions should address the specific points made by each reviewer.

(1) A letter containing a detailed list of your responses to the review comments and a description of the changes you have made in the manuscript. Please note while forming your response, if your article is accepted, you may have the opportunity to make the peer review history publicly available. The record will include editor decision letters (with reviews) and your responses to reviewer comments. If eligible, we will contact you to opt in or out.

(2) Two versions of the manuscript: one with either highlights or tracked changes denoting where the text has been changed; the other a clean version (uploaded as the manuscript file).

Additionally, to enhance the reproducibility of your results, PLOS recommends that you deposit your laboratory protocols in protocols.io, where a protocol can be assigned its own identifier (DOI) such that it can be cited independently in the future. For instructions see http://journals.plos.org/plospathogens/s/submission-guidelines#loc-materials-and-methods

We hope to receive your revised manuscript within 60 days. If you anticipate any delay in its return, we ask that you let us know the expected resubmission date by replying to this email. Revised manuscripts received beyond 60 days may require evaluation and peer review similar to that applied to newly submitted manuscripts.

[LINK]

Sincerely,

William C Gause

Associate Editor

PLOS Pathogens

James Kazura

Section Editor

PLOS Pathogens

Kasturi Haldar

Editor-in-Chief

PLOS Pathogens

orcid.org/0000-0001-5065-158X

Grant McFadden

Editor-in-Chief

PLOS Pathogens

orcid.org/0000-0002-2556-3526

Reviewer's Responses to Questions

**Part I - Summary**

Reviewer #1: Animal infection experiments are usually carried out with single bolus inoculations, whereas natural infections often involve repeat exposures to low doses. In this manuscript, the authors examine the outcome of repetitive low dose inoculations by gastrointestinal helminths using trickle infection of mice by Trichuris muris as a model. Their main finding is that the initial response is dominated by Th1, which eventually switches to a Th2 response that leads to worm expulsion. Although the outcome of trickle infection did not reveal any surprising aspect of parasite immunity, this study will be an important resource to the field as it demonstrates that repetitive low doses evokes a response that is distinct from previous findings in which mice receive one low dose that leads to a chronic infection. This study establishes the model and carefully documents the time course of infection. There are a few concerns regarding interpretation of the results that the authors may want to address prior to disseminating the findings to the field.

Reviewer #2: In this study presented by Glover et al. the authors seek to investigate how immunity to T. muris changes following repeat exposures to low doses of parasites. This is an extremely important question and one that is not well understood considering that the majority of studies investigating immunity to helminths focus on determining how the host responds to a single large dose of parasites. The model that is being employed in this study is much more indicative of what is occurring in human patients and has the potential substantially inform our understanding of how immunity is truly developed.

By using this system, the authors show that there is a transition point at week 11 at which point the mice develop immunity and start to clear the worms. They nicely demonstrate that this transition appears to coincide with increase production of IL-13, mucus production and goblet cell responses. Further, they take a loss-of-function approach and show that CD4+ T cells appear to be the main mediators of this acquired protection. Further, they demonstrate that acquired immunity following a trickle infection is maintained and protective following a subsequent challenge. Finally, they show that the composition of microbial communities in the gut is altered in mice that remain susceptible to infection, but begins to return to baseline as immunity it acquired.

Reviewer #3: The study by Glover et al presents a very thorough and clear report of how trickle infection with Trichuris impacts on host resistance and immunity. The rationale for the study is that the bulk of the literature utilities bolus infections (one or more applications of large egg/larval numbers to the host) whereas trickle infection represents the nature case (for all animals, including rodents that can be infected with Trichuris) and that it would therefore be important to determine the impact of trickle infection on host responsiveness and worm burdens.

It is commendable that the authors took the time to properly address these largely ignored but important issue, and I have little doubt that their efforts will help both inform and alter the work of all scientists interesting in modeling host-helminth interactions.

As a whole the work is concisely presented and represents a large and well controlled body of data. The only comments I have for improvement of the manuscript relate to data inclusion and/or presentation and editorial modifications as can be found below.

**Part II – Major Issues: Key Experiments Required for Acceptance**

Reviewer #1: There are 3 variables that change when the authors compare early time points versus the late time points – age of the mice, number of inoculations, and the amount of time that has passed since the first inoculation. Therefore, it is difficult to know what led to alterations in worm burden and Th responses at the later time points. What happens if they perform repetitive inoculations for the first 3-5 weeks and then examine at week 11?

Reviewer #2: As stated above, careful studies of this nature are extremely important and are likely to be very informative. Although the authors have nicely characterized the model and the dramatic changes that are occurring between weeks 9-11, there is no explanation of the mechanisms that drive this transition. In its current state, the study is very observational. There appears to be an important transition in the Th2 cells that occurs after week 9 despite the fact that by week 9 the mice have been challenged 7 times and have 50 worms. Why do 2 more challenges drive this difference, is it just a magnitude issue? What do the authors think is happening, considering that little change in the worm burden is occuring between weeks 7-9 it doesn’t seem like its simply antigen load. It would be great if they authors could determine whether T cells transferred on week 9 and 11 differed in their ability to promote immunity using a gain of function approach. They should them determine what has truly changes in those T cells to give mechanistic insight. Transcriptional profiling would be ideal to accomplish this assuming that its T cell intrinsic. In general, the manuscript needs more mechanistic data to be appropriate for publication.

Also, although interesting the microbiome analysis seems out of place and is very observational in its current form. What are the authors trying to say about these findings and how they relate to the development of immunity?

Reviewer #3: none

**Part III – Minor Issues: Editorial and Data Presentation Modifications**

Reviewer #1: In Fig 5b, have the authors confirmed that the dose of anthelmintic used was effective at clearing worms?

For Fig 4, please specify the time points and egg doses in the legend

Reviewer #2: (No Response)

Reviewer #3: Comments related to data and figure legends:

1) The experimental design of the work presented Figure 5 was extremely interesting and is given a strong degree of relevance by the authors in that a entire subsection (lines 160--174) is detected to it. However a very limited data set is shown and as such the data stands in stark contrast to the rest of the manuscript. Many questions remain open including the impact on specific larval stages, the association with various cytokines and antibodies, epithelial cell responses and so forth.

For these reasons I would recommend one of the following options i) extending the dataset, ii) moving it to the supplementary information, or iii) removing this figure and instead working further to determine whether ILC2 or CD4 T cells were involved in the long-lasting protection observed, and to determine the functional significance of antibodies (perhaps by passive transfer exps) at this time.

2) It is not clear whether the data shown in Figure 9 is pooled from two experiments or is one data set that is representative of 2 exps. Given the worm counts are very variable and the possible trend toward increased worm counts in the DTx treated mice it would be useful to either show the second experiment or to increase the total number of mice investigated to increase power.

3) The pictures of tuft cells shown in Fig8 are not convincing. Better representative pictures and/or pictures at higher magnification should be shown to convince the readers the authors performed a correct quantification of tuft cells. It might also be useful to show pictures from the SI of Nippo infected mice for comparison and again to convince the readers correct techniques were employed in enumerating tuft cells.

Comments related to editoral changes:

1) many of the figure legends do not contain enough information to allow the reader to understand the experiment, this is especially true of the supplementary figures. For instance no information is given as to the length or timing of the Edu pulse (figures 4 and 7 or how antibody levels were quantified (Supplementary)

2) It would be useful for the authors to clearly state in the text that the microbiota analyses was performed in stool, and to discuss the drawbacks of using stool versus cecal contents and mucous layers in terms of exploring the true extent of the impact of infection.

3) I am not sure I agree with the authors conclusion that antibody levels were unaffected by CD4 T cell depletion (they seem to be decreased)? As no evidence regarding the role of antibodies is provided it would be better to refrain from raising their involvement as being unlikely and instead simply say that this was not determined.

PLOS authors have the option to publish the peer review history of their article (what does this mean?). If published, this will include your full peer review and any attached files.

Reviewer #1: No

Reviewer #2: No

Reviewer #3: No

---

## [Decision Letter · Decision Letter 1]

9 Oct 2019

Dear Prof. Grencis:

Thank you very much for submitting your manuscript "Trickle infection and immunity to Trichuris muris" (PPATHOGENS-D-19-01054R1) for review by PLOS Pathogens. Your manuscript was fully evaluated at the editorial level and by independent peer reviewers. The reviewers appreciated the attention to an important topic but identified some aspects of the manuscript that should be improved.

We therefore ask you to modify the manuscript according to the review recommendations before we can consider your manuscript for acceptance. Your revisions should address the specific points made by each reviewer.

(1) A letter containing a detailed list of your responses to the review comments and a description of the changes you have made in the manuscript. Please note while forming your response, if your article is accepted, you may have the opportunity to make the peer review history publicly available. The record will include editor decision letters (with reviews) and your responses to reviewer comments. If eligible, we will contact you to opt in or out.

(2) Two versions of the manuscript: one with either highlights or tracked changes denoting where the text has been changed; the other a clean version (uploaded as the manuscript file).

We hope to receive your revised manuscript within 60 days or less. If you anticipate any delay in its return, we ask that you let us know the expected resubmission date by replying to this email.

[LINK]

Sincerely,

William C Gause

Associate Editor

PLOS Pathogens

James Kazura

Section Editor

PLOS Pathogens

Kasturi Haldar

Editor-in-Chief

PLOS Pathogens

orcid.org/0000-0001-5065-158X

Grant McFadden

Editor-in-Chief

PLOS Pathogens

orcid.org/0000-0002-2556-3526

Reviewer's Responses to Questions

**Part I - Summary**

Reviewer #1: The authors have used the literature to argue that the control conditions for their experiments have been performed extensively by other groups and that their observations cannot be explained by confounding variables such as the age of the mice and elapsed time. I accept these points and also appreciate the responses to the minor issues that were raised. I look forward to seeing the publication of this important study.

Reviewer #2: As stated perviously, I believe this study is of great importance and will help advance the field if some more mechanistic data can be presented. The authors addressed some of my concerns with the adoptive transfer studies they presented, but I believe they are very important and should be added to the manuscript. It is entirely possible that the T cells are necessary but not sufficient for the changes in immunity. This suggest that T cells are the main effector cells but their actions are being influenced by increased mediators of activation or decreased suppressive mechanisms. I believe it is entirely within the scope of the study to better define these pathways. At the least, can the authors confirm that T cells are necessary but not sufficient and then postulate why that may be the case. Although I would like to see them take a deeper dive into the mechanisms I am very enthusiastic about this work and believe that level of depth would be very informative for the design of future studies and greatly inform the field.

Reviewer #3: The authors have addressed all the authors comments adequately

**Part II – Major Issues: Key Experiments Required for Acceptance**

Reviewer #1: (No Response)

Reviewer #2: (No Response)

Reviewer #3: (No Response)

**Part III – Minor Issues: Editorial and Data Presentation Modifications**

Reviewer #1: (No Response)

Reviewer #2: (No Response)

Reviewer #3: (No Response)

PLOS authors have the option to publish the peer review history of their article (what does this mean?). If published, this will include your full peer review and any attached files.

Reviewer #1: No

Reviewer #2: No

Reviewer #3: No

---

## [Editor Report · Decision Letter 2]

29 Oct 2019

Dear Prof. Grencis,

We are pleased to inform that your manuscript, "Trickle infection and immunity to Trichuris muris", has been editorially accepted for publication at PLOS Pathogens. 

Before your manuscript can be formally accepted and sent to production, you will need to complete our formatting changes, which you will receive by email within a week. Please note that your manuscript will not be scheduled for publication until you have made the required changes.

IMPORTANT NOTES

(1) Please note, once your paper is accepted, an uncorrected proof of your manuscript will be published online ahead of the final version, unless you’ve already opted out via the online submission form. If, for any reason, you do not want an earlier version of your manuscript published online or are unsure if you have already indicated as such, please let the journal staff know immediately at plospathogens@plos.org.

(2) Copyediting and Proofreading: The corresponding author will receive a typeset proof for review, to ensure errors have not been introduced during production. Please review the PDF proof of your manuscript carefully, as this is the last chance to correct any errors. Please note that major changes, or those which affect the scientific understanding of the work, will likely cause delays to the publication date of your manuscript. 

(3) Appropriate Figure Files: Please remove all name and figure # text from your figure files. Please also take this time to check that your figures are of high resolution, which will improve the readbility of your figures and help expedite your manuscript's publication. Please note that figures must have been originally created at 300dpi or higher. Do not manually increase the resolution of your files. For instructions on how to properly obtain high quality images, please review our Figure Guidelines, with examples at: http://journals.plos.org/plospathogens/s/figures.

(4) Striking Image: Please upload a striking still image to accompany your article if one is available (you can include a new image or an existing one from within your manuscript). Should your paper be accepted, this image will be considered for our monthly issue image and may also appear on our website to feature your article. Please upload this as a separate file, selecting "striking image" as the file type upon upload. Please also include a separate "Other" file with a caption, including credits and any potential copyright information. Please do not include the caption in the main article file. If your image is from someone other than yourself, please ensure that the artist has read and agreed to the terms and conditions of the Creative Commons Attribution License at http://journals.plos.org/plospathogens/s/content-license. Please note that PLOS cannot publish copyrighted images.

(5) Press Release or Related Media: If your institution or institutions have a press office, please notify them about your upcoming paper at this point, to enable them to help maximize its impact. If they will be preparing press materials for this manuscript, please inform our press team in advance at plospathogens@plos.org as soon as possible. We ask that you contact us within one week to plan ahead of our fast Production schedule. If you need to know your paper's publication date for related media purposes, you must coordinate with our press team, and your manuscript will remain under a strict press embargo until the publication date and time. This means an early version of your manuscript will not be published ahead of your final version. 

(6)  PLOS requires an ORCID iD for all corresponding authors on papers submitted after December 6th, 2016. Please ensure that you have an ORCID iD and that it is validated in Editorial Manager.  To do this, go to ‘Update my Information’ (in the upper left-hand corner of the main menu), and click on the Fetch/Validate link next to the ORCID field.  This will take you to the ORCID site and allow you to create a new iD or authenticate a pre-existing iD in Editorial Manager

(7) Update your Profile Information: Now that your manuscript has been provisionally accepted, please log into Editorial Manager and update your profile, if needed. Go to https://www.editorialmanager.com/ppathogens, log in, and click on the "Update My Information" link at the top of the page. Please update your user information to ensure an efficient production and billing process. 

(8) LaTeX users only: Our staff will ask you to upload a TEX file in addition to the PDF before the paper can be sent to typesetting, so please carefully review our Latex Guidelines http://journals.plos.org/plospathogens/s/latex in the meantime.

(9) If you have associated protocols in protocols.io, please ensure that you make them public before publication to guarantee immediate access to the methodological details.

Best regards,

William C Gause

Associate Editor

PLOS Pathogens

James Kazura

Section Editor

PLOS Pathogens

Kasturi Haldar

Editor-in-Chief

PLOS Pathogens

orcid.org/0000-0001-5065-158X

Grant McFadden

Editor-in-Chief

PLOS Pathogens

orcid.org/0000-0002-2556-3526
---

## [Editor Report · Acceptance letter]

5 Nov 2019

Dear Prof. Grencis,

We are delighted to inform you that your manuscript, "Trickle infection and immunity to Trichuris muris," has been formally accepted for publication in PLOS Pathogens.

Best regards,

Kasturi Haldar

Editor-in-Chief

PLOS Pathogens

orcid.org/0000-0001-5065-158X

Grant McFadden

Editor-in-Chief

PLOS Pathogens

orcid.org/0000-0002-2556-3526